# Physics of Stratocumulus Top (POST): turbulence characteristics

**I. Jen-La Plante[1], Y-F. Ma[1], K. Nurowska[1], H. Gerber[2], D. Khelif[3], K. Karpinska[1], M. K. Kopec[1], W. Kumala[1], and S. P. Malinowski[1]**

[1]Institute of Geophysics, Faculty of Physics, University of Warsaw, Poland
[2]Gerber Scientific Inc., Reston, VA, USA
[3]Department of Mechanical and Aerospace Engineering, University of California, Irvine, CA, USA

*Correspondence to:* Szymon P. Malinowski
malina@fuw.edu.pl

**Abstract.** Turbulence observed during the Physics of Stratocumulus Top (POST) research campaign is analyzed. Using in-flight measurements of dynamic and thermodynamic variables at the interface between the stratocumulus cloud top and free troposphere, the cloud top region is classified into sublayers, and the thicknesses of these sublayers are estimated. The data are used to calculate turbulence characteristics, including the bulk Richardson number, mean-square velocity fluctuations, turbulence kinetic energy (TKE), TKE dissipation rate, and Corrsin, Ozmidov and Kolmogorov scales. A comparison of these properties among different sublayers indicates that the entrainment interfacial layer consists of two significantly different sublayers: the turbulent inversion sublayer (TISL) and the moist, yet hydrostatically stable, cloud top mixing sublayer (CTMSL). Both sublayers are marginally turbulent, i.e. the bulk Richardson number across the layers is critical. This means that turbulence is produced by shear and damped by buoyancy such that the sublayer thicknesses adapt to temperature and wind variations across them. Turbulence in both sublayers is anisotropic, with Corrsin and Ozmidov scales as small as $\sim 30cm$ and $\sim 3m$ in the TISL and CTMSL, respectively. These values are $\sim 60$ and $\sim 15$ times smaller than typical layer depths, indicating flattened large eddies and suggesting no direct mixing of cloud top and free tropospheric air. Also, small scales of turbulence are different in sublayers as indicated by the corresponding values of Kolmogorov scales and buoyant and shear Reynolds numbers.

## 1 Introduction

Turbulence is a key cloud process governing entrainment and mixing, influencing droplet collisions, and interacting with large-scale cloud dynamics. It is unevenly distributed over time and space due to its inherent intermittent nature as well as various sources and sinks changing during the cloud life cycle (Bodenschatz et al., 2010). Turbulence is difficult to measure. Reports on the characterization of cloud-related turbulence based on in situ data are scarce in the literature (see, e.g., the discussion in Devenish et al. (2012)). This study aims to characterize stationary or slowly changing turbulence in a geometrically simple yet meteorologically important cloud-clear air interface at the top of marine stratocumulus.

Characterization of stratocumulus top turbulence is interesting for a number of reasons, including our deficient understanding of the entrainment process (see, e.g., Wood (2012)). Typical stratocumulus clouds are shallow and have low liquid water content (LWC). Such clouds are sensitive to mixing with dry and warm air from above, which may lead to cloud top entrainment instability and thus cloud dissipation according to theory (Deardorff , 1980; Randall, 1980). However, the theory based on thermodynamic analysis only is not sufficient. For instance Kuo and Schubert (1988) and recently Stevens (2010) and van der Dussen et al. (2014) argued that stratocumulus clouds often persist while being within the buoyancy reversal regime. Turbulent transport across the inversion is a mechanism that governs exchange between the cloud top and free atmosphere and should be considered.

Convection in the stratocumulus topped boundary layer (STBL) is limited. Updrafts in the STBL, in contrast to those in the diurnal convective layer over ground, do not

penetrate the inversion (see, e.g., the LES simulations by Kurowski et al. (2009) and analysis in Haman (2009)). Such updrafts, diverging below the hydrostatically stable layer, may contribute to turbulence just below and within the inversion. Researchers have known for years (e.g., Brost et al. (1982)) that wind shear in and above the cloud top is important or even dominating source of turbulence in this region. Finally, radiative and evaporative cooling can also produce turbulence by buoyancy fluctuations. These multiple sources are responsible for exchange across the inversion.

There is experimental evidence that mixing at the stratocumulus top leads to the formation of a specific layer, called the entrainment interfacial layer (EIL) after Caughey et al. (1982). Several airborne research campaigns were aimed at investigating stratocumulus cloud top dynamics and thus the properties of the EIL, such as DYCOMS (Lenshow et al., 1988) and DYCOMS II (Stevens et al., 2003). The results (see, e.g., Lenshow et al. (2000); Gerber et al. (2005); Haman et al. (2007)) indicate the presence of turbulence in the EIL, including inversion capping the STBL. Ongoing turbulent mixing generates complex patterns of temperature and liquid water content at the cloud top. The EIL is typically relatively thin and uneven (thickness of few tens of meters, fluctuating from single meters to ∼100m). Many numerical simulations based on RF01 of DYCOMS II (e.g., Stevens et al. (2005); Moeng et al. (2005); Kurowski et al. (2009)) confirm that the cloud top region is characterized by the intensive production of turbulence kinetic energy (TKE) and turbulence in the EIL.

Recently, airborne measurements of fine spatial resolution (at the centimeter scale for some parameters), aimed at providing a better understanding of the EIL, were performed in the course of Physics of Stratocumulus Top (POST) field campaign (Gerber et al., 2010, 2013; Carman et al., 2012). A large dataset was collected from sampling the marine stratocumulus top during porpoising (flying with a rising and falling motion) across the EIL and is freely available for analysis (see http://www.eol.ucar.edu/projects/post/). An analysis of the POST data by Gerber et al. (2013) confirmed that the EIL is thin, turbulent and of variable thickness. This result is in agreement with measurements by Katzwinkel et al. (2011), performed with a helicopter-borne instrumental platform penetrating the inversion capping the stratocumulus. These measurements indicated that the uppermost cloud layer and capping inversion are turbulent and that wind shear across the EIL is a source of this turbulence. Malinowski et al. (2013) confirmed the role of wind shear using data from two thermodynamically different flights of POST. They also proposed an empirically based division of the stratocumulus top region into sublayers based on the vertical profiles of wind shear, stability and the thermodynamic properties of the air. An analysis of the dynamic stability of the EIL using the gradient Richardson number $R_i$ confirmed the hypothesis presented by Wang et al. (2008, 2012) and Katzwinkel et al. (2011) that the thickness of the turbu-

lent EIL changes based on meteorological conditions (temperature and wind variations between the cloud top and free troposphere) such that the Richardson number across the EIL and its sublayers is close to the critical value.

In the present paper, we begin from extension of the analysis of the POST data by Malinowski et al. (2013) to a larger number of cases. Then, we discuss performance of the algorithmic layer division, allowing for objective distinction of cloud top sublayers. As a main part of the study we analyze the properties of turbulence in the sublayers to provide detailed characterization of turbulence in the stratocumulus cloud top region, based on a wide range of measurement data. Finally, we discuss the consequences of the fine structure of the turbulent cloud top and capping inversion, with a focus on the vertical variability of turbulence and characteristic length scales.

## 2   Data and Methods

The POST experiment collected in situ measurements of thermodynamic and dynamic variables at the interface between the stratocumulus cloud top and free troposphere in a series of research flights near Monterey Bay ($\sim 100km$ south from San Francisco, California) during July and August 2008. The CIRPAS Twin Otter research aircraft was equipped to measure temperature with a resolution down to the centimeter scale (Kumala et al., 2013), LWC with a resolution of $\sim 5cm$ (Gerber et al., 1994), humidity and turbulence with a resolution of $\sim 1.5m$ (Khelif et al., 1999), as well as short- and longwave radiation, aerosol and cloud microphysics. To study the vertical structure of the EIL, the flight pattern consisted of shallow porpoises ascending and descending through the cloud top at a rate of $1.5m/s$ flying with a true airspeed of $\sim 55m/s$. The flight profiles indicating the data collection strategy are presented in Fig.1. Details of the apparatus and observations are provided in Gerber et al. (2010); Carman et al. (2012); Gerber et al. (2013). Meteorological conditions in the course of the measurements were stable in the Eastern North Pacific high pressure area with cloud tops were located between $375m$ and $760m$ (mean is $513 \pm 137m$), stable wind direction (between 320 and 340 degrees) and speeds ($6.5 - 14.5m/s$) at the cloud top height, with the wind shear (sometimes directional) above cloud tops. Typical temperature at the cloud top was $10.8^oC$, temperature jumps across the inversion varied in a range $2.3 - 10.2K$. More details concerning conditions in the course of flights can be found in tables 1-4 of Gerber et al. (2013) and in the open POST database (http://www.eol.ucar.edu/projects/post/).

The 15 measurement flights of POST were originally divided by Gerber et al. (2010) into two categories, described as "classical" and "non-classical". Examples from each category, classical flight TO10 and non-classical flight TO13, closely examined in Malinowski et al. (2013), are also in-

cluded in this study. The original classification by Gerber was based on correlation of LWC and vertical velocity fluctuations in diluted cloud volumes, but Malinowski et al. (2013) found that classical cases exhibit monotonic increases in LWC with altitude across the cloud depth, sharp, shallow and strong capping inversion, and dry air in the free troposphere above. Non-classical cases are characterized by LWC fluctuations in the upper part of the cloud, weaker inversion, more temperature fluctuations in the cloud top region as well as more humid air above the inversion. A more detailed analysis of all POST flights indicated that the division into these categories is not straightforward and that a wide variety of cloud top behaviors spanning the entire spectrum between "classical" and "non-classical" regimes can be found.

The present study extends the analysis of two extreme "classical" and "non-classical" cases performed by Malinowski et al. (2013) to more flights from the POST data set. Using Tables 1, 2 and 4 of Gerber et al. (2013) from all 17 POST flight we selected 8 cases (TO03, TO05, TO06, TO07, TO10, TO12, TO13, TO14), which cover the whole range of observed temperature and humidity jumps across the inversion, shear strengths, cloud top change rates, entrainment velocities, buoyancies of cloud-clear air mixtures and day/night conditions (c.f. Tab.1 for key parameters). For these cases we repeated analyses of Malinowski et al. (2013) performing layer division, and estimating Richardson Numbers across the layers. Then, in order to understand dynamics of mixing process, we determined turbulence characteristics in the layers. We used measurements of three components of wind velocity and fluctuations, sampled at a rate of 40 Hz with a five-hole gust probe and corrected for the motion of the aircraft (Khelif et al., 1999). We estimated values of Turbulence Kinetic Energy (TKE) and velocity variances in the layers, TKE dissipation rates, and finally, characterized anisotropy of turbulence.

## 2.1 Layer division

Systematic and repeatable changes in the dynamic and thermodynamic properties of the air observed in the porpoising flight pattern allowed for the introduction of an algorithmic division of the cloud top region into sublayers, as illustrated in Fig.1. In brief, the method identifies the vertical divisions between the stable free troposphere (FT) above the cloud, the EIL consisting of a turbulent inversion sublayer (TISL) characterized by temperature inversion and wind shear, and of a moist and sheared cloud top mixing sublayer (CTMSL), and, finally, the well-mixed cloud top layer (CTL)

The classification method is described in detail in Malinowski et al. (2013) and summarized here. First, the division between the FT and TISL is identified by the highest point where the gradient of liquid water potential temperature exceeds $0.2 K/m$ and the turbulence kinetic energy (TKE) exceeds 0.01 m$^2$/s$^2$. Next, the division between the TISL and CTMSL corresponds to the uppermost point where LWC

exceeds 0.05 g/m$^3$. The final division between the CTMSL and CTL is determined by the point at which the square of the horizontal wind shear reaches 90% of the maximum, usually collocated with the location where the remarkable temperature fluctuations disappear. For graphical examples of cloud top penetration and the layer division, see Figs. 4, 5, 12 and 13 in Malinowski et al. (2013).

We applied the layer division algorithm to POST flights TO3, TO5, TO6, TO7, TO10, TO12, TO13 and TO14 to all ascending/descending segments of the flight. Points separating FT from TISL, TISL from CTMSL and CTMSL from CTL were found in most cases. Sometimes either division between FT and TISL or division between CTMSL and CTL was not detected. This was most probably a result of too shallow individual porpoises. Before the experiment, in the course of discussion of flight pattern, it was decided that porpoises should be within a range of $sim100$ m from the cloud top. Actual decision to stop ascent or descent was taken by the pilot based on this recommendation. A posteriori, in seems that sometimes slightly deeper porpoises would be more appropriate. Division algorithm, proposed on a basis of the available data, disregarded division points detected too close to the local extremum of the aircraft altitude in order to avoid false estimates of the wind shear (division CTMSL/CTL) and TKE or temperature gradient (FT-TISL).

The example effect of the division algorithm is plotted in Fig.1, while all results, together with additional information about flights are summarized in Tab.1. In total, the layer division applied to 8 different stratocumulus cases, resulted in the successful definition of sublayers in 17-58 cloud top penetrations for each case. Such a rich data set allows for a comprehensive description of the cloud top structure and turbulence properties across the EIL, its sublayers and adjacent layers of the FT and CTL.

In order to illustrate the rationale for the layer division in Fig.2 we present two randomly selected cloud penetrations from "non-classical" TO5 and "classical" TO12 cases (another examples can be found in Malinowski et al. (2013)). Wind shear across the whole EIL present in both cases, usually weaker across CTMSL than across TISL. Wind velocity fluctuations in TISL are less significant than in CTMSL. TISL is characterized by large mean temeperature gradient (high static stability) and remarkable temperature fluctuations in dry environment. In CTMSL only a weak mean temperature gradient is present, temperature fluctuations are small, but the layer is moist and LWC rapidly fluctuates between the maximum value for cloud and zero. Such striking differences indicate that division of the EIL into two sublayers is fully justified. But another question may arise: is division between CTMSL and CTL justified? The answer is yes, and the first part of the proof is in Malinowski et al. (2013), who show that turbulence in CTMSL is marginal in terms of Richardson number analysis. For more arguments behind this division let's investigate turbulence in both sublayers and adjacent FT and CTL.

In order to characterize turbulence, Reynolds decomposition must be used for the mean and turbulent velocity components. In atmospheric conditions, important assumptions of rigorous decomposition (e.g., averaging on the entire statistical ensemble of velocities) are not fulfilled, and averaging is often performed on short time series. Specific problems related to the averaging of POST airborne data result from the layered structure of the stratocumulus top region and porpoising flight pattern. The main issue is determining how to average collected data to reasonably estimate the mean and fluctuating quantities in all layers. The assumptions are that layers are reasonably uniform (in terms of turbulence statistics) and that averaging must be performed on several (the more the better) large eddies. At a true aircraft airspeed of $55m/s$, an ascent/descent velocity of 1.5 m/s and a sampling rate of 40 Hz over 300 data points corresponds to a distance of $\sim 410m$ in the horizontal direction and of $\sim 11m$ in the vertical direction. Assuming the characteristic horizontal size of large eddies of the order of $\sim 100m$, such averaging accounts for 3–5 large eddies and captures the fine structure of the cloud top with a resolution of $\sim 10m$ in the vertical direction. This resolution should be sufficient based on estimates of the EIL thickness by Haman et al. (2007) and Kurowski et al. (2009) and noting that their definition of the EIL corresponds to the TISL in the present study. To illustrate the effect of averaging in Fig.2, the averaged (centered running mean on 300 points) values of all three velocity components are plotted. Tests on various porpoises from all investigated research flights using averaging lengths varying from 100 to 500 points and different techniques (centered running mean, segment averaging) confirmed that the proposed approach applied to POST data gives results that allow the layers to be distinguished and statistics sufficient to characterize the turbulent fluctuations within each layer to be obtained.

## 3    Analysis

### 3.1    Thickness of the sublayers

The results in Tab.1 indicate that for all flights, the depth of the TISL is smaller than that of the CTMSL. The thicknesses of the sublayers vary from $\sim 10m$ to $\sim 100m$, in accordance with the aforementioned studies. The relatively large standard deviation of the layer thickness prevents general conclusions from being made. The only exception concerns cases classified as "classical" and, according to the analysis in Gerber et al. (2013), permitting for the potential production of a negatively buoyant mixture of cloud top and free tropospheric air in the adiabatic process. These TO6, TO10 and TO12 flights generated the thinnest CTMSL, in agreement with the schematic of the EIL structure proposed by Malinowski et al. (2013) (see Fig. 16 therein), who argued that thickness of the CTMSL diminishes with growing CTEI. Similar structure of "classical" non-POST stratocumulus was

also reported in numerical simulations of CTEI permitting in the DYCOMS RF01 case by Mellado et al. (2014), who demonstrated a "peeling off" of the negatively buoyant volumes from the shear layer at the cloud top.

### 3.2    Bulk Richardson Number

To compare the newly processed flights with TO10 and TO13 discussed in Malinowski et al. (2013), we analyze the bulk Richardson numbers of the porpoises using the same procedure (c.f. sections 4.1 and 4.2 therein). Briefly, averaging and layer division allowed for the estimation of $R_i$ using the following formula:

$$R_i = \frac{\frac{g}{\theta}\left(\frac{\Delta\theta}{\Delta z}\right)}{\left(\frac{\Delta u}{\Delta z}\right)^2 + \left(\frac{\Delta v}{\Delta z}\right)^2}. \tag{1}$$

Here, $g$ is the acceleration due to gravity and $\Delta\theta$, $\Delta u$ and $\Delta v$ are the jumps of virtual potential temperature and horizontal velocity components across the depth of the layer $\Delta z$.

The resulting histograms of the bulk Richardson number, $R_i$, from flight segments across the consecutive layers (FT, TISL, CTMSL and CTL) as well as the EIL, defined as TISL+CTMSL, for all investigated cases are summarized in Fig.3.

Prevailing $R_i$ estimates in FT indicate turbulence damped by static stability, i.e., $R_i > 1$ (Grachev et al., 2012). For presentation purposes, several extremely high values of $R_i$ measured are not presented in these figures. The $R_i$ estimates in the TISL and CTMSL indicate the prevailing marginal turbulence neutral stability across these layers (i.e., $0.75 \gtrsim R_i \gtrsim 0.25$ dominate). Interestingly, the $R_i$ distributions for "classical" cases TO6, TO10 and TO12 show long positive tails in the CTMSL. Below, in the CTL, dominating bins document a neutral stability or weak convective instability, as expected within the STBL.

The positive tails of the $R_i$ distributions in the FT and CTL are partially due to the fact that the vertical gradients of the horizontal velocity components are small in these layers, i.e., the denominator in the $R_i$ definition is close to zero. Division by a near-zero value does not occur in the CTMSL, and values of $R_i > 0.75$ indicate that the layer was dynamically stable on these porpoises. This suggests an intermittent structure of the layer, e.g., the coexistence of intense turbulence patches and regions of decaying or even negligible turbulence.

In summary, the results of the $R_i$ analysis for the new flights are in agreement with those of Malinowski et al. (2013), confirming that the thickness of the EIL sublayers $\Delta Z$,

$$\Delta Z = R_{iC}\left(\frac{\theta}{g}\right)\left(\frac{\Delta u^2 + \Delta v^2}{\Delta\theta}\right) \tag{2}$$

is such that $R_i$ across them is close to the critical value, i.e., in the range $0.75 \gtrless R_{iC} \gtrless 0.25$.

The above relation is equivalent to Eq. 6 in Mellado et al. (2014), who analyze the results of numerical simulations of stratocumulus top mixing and adopted estimates of the asymptotic thickness of shear layers in oceanic flows (Smyth and Moum, 2000; Brucker and Sarkar, 2007) and in the cloud-free atmospheric boundary layer (Conzemius and Fedorovich , 2007).

### 3.3 Turbulent Kinetic Energy (TKE)

Adopting the averaging procedure allows for the characterization of the RMS (Root Mean Square) fluctuations of all three components of velocity in the cloud top sublayers as well as the mean kinetic energy:

$$\text{TKE} = \frac{1}{2}(\overline{u'^2} + \overline{v'^2} + \overline{w'^2}). \tag{3}$$

In the above, $u'$, $v'$, and $w'$ are fluctuations of the velocity components calculated using a 300-point averaging window to establish the mean value of velocity (Sec. 2.2) and averaging of these fluctuations across the layer depth and on all suitable porpoises for a given flight. The results are shown in Table 2 and graphically presented in Fig.4.

An analysis of the results illustrates two important properties of turbulence:

1) the anisotropy of turbulence in the TISL and CTMSL, revealed by reduced velocity fluctuations in the vertical direction (compared to the horizontal direction)

2) the presence of the maximum TKE in the CTMSL (in the majority of cases).

TO13 is the only flight showing larger vertical than horizontal velocity fluctuations in the TISL. However, this flight is characterized by the weakest inversion (Gerber et al., 2013), nearly thinnest TISL (Tab.1) and largest vertical velocity fluctuations in the FT. This suggests that the non-typical picture of vertical velocity fluctuations results from the presence of gravity waves, which substantially modify the vertical velocity variance just above the cloud top. This hypothesis is supported by the observations of an on-board scientist (flight notes are available in the POST database), who wrote: "Cloud tops looked like moguls". Numerical simulations of the TO13 case suggest the presence of gravity waves at and above the inversion.

For many flights, in the CTL, where the Richardson number suggests the production of turbulence due to static instability, there are weak signatures on the opposite anisotropy than in the layers above, i.e., the vertical velocity fluctuations exceed the horizontal ones.

### 3.4 TKE dissipation rate

Derivation of the TKE dissipation rate from moderate-resolution airborne measurements is always problematic.

The assumptions of isotropy, homogeneity and stationarity of turbulence, used to calculate the mean TKE dissipation rate from power spectra and/or structure functions, are hardy, if ever, fulfilled. This is also the case in our investigation of highly variable thin sublayers of the STBL top and is enhanced by the porpoising flight pattern. Considering these problems, we estimated the TKE dissipation rate by two methods. Three spatial components of velocity fluctuations are treated separately, allowing for the study of possible anisotropy, which is expected due to the different stability and shear in the stratocumulus top sublayers.

#### 3.4.1 Estimates from the power spectral density

The first method was to estimate the TKE dissipation rate $\varepsilon$ using power spectral density (PSD) of turbulence fluctuations in a similar manner as, e.g., Siebert et al. (2006):

$$P(f) = \alpha \overline{\varepsilon}^{2/3} \left(\frac{\overline{U}}{2\pi}\right)^{\frac{2}{3}} f^{\frac{-5}{3}} \tag{4}$$

where $\overline{U}$ is the average speed of the plane, $f$ is the frequency, $P(f)$ is the power spectrum of velocity fluctuations, and $\alpha$ is the one-dimensional Kolmogorov constant, with a value of 0.5. On a logarithmic scale, the spectrum should be described by a line with a slope of $-5/3$ as a function of frequency. $\varepsilon$ can be estimated by fitting the $-5/3$ line in the log-log plot.

Originally, the relationship assumes local isotropy, stationarity and horizontal homogeneity of turbulence. The first assumption, as indicated by the analysis of velocity fluctuations, is not fulfilled. To investigate this problem in more detail, we analyse spectra for all three components independently. Stationarity and horizontal homogeneity are accounted for constructing composite PSDs for each layer by summing individual PSDs for all suitable penetrations.

Power spectrum from penetration through the investigated layer, $P(f)$, is calculated using the Welch method in MATLAB with a moving window of $2^8$ points on the 40 Hz velocity data. This is done individually for each component of the velocity. The fluctuations are determined with respect to a moving average of 300 points, as in the layer division. Then each velocity spectrum fulfilling the quality criterion for each velocity component is combined into a composite spectrum for every flight. Finally the $-5/3$ line is fitted in log-log coordinates. Figure 5 shows all the composite power spectra on a logarithmic scale, with the three velocity components spread out by factors of 10. The line with a slope $-5/3$ indicated by equation 4 is shown by the dashed line fits in the figure. The fit is limited to the frequency range of $0.3 - 5Hz$, neglecting the higher frequency features attributed to interactions with the plane (and the lower frequency artefacts of the Welch method). The spectra in the CTMSL and CTL correspond well with the $-5/3$ law in the analyzed range of scales. A small amplitude decrease of vertical velocity fluctuations

at frequencies below $0.3 - 1Hz$ (depending on the flight) can be observed in the CTMSL. In the TISL, the scaling of velocity fluctuations with the $-5/3$ law is less evident; various deviations from a constant slope are more evident in some flights (TO03, TO07, TO10, TO13) than in others. In the FT, scaling is poor; specifically, the spectra are steeper than $-5/3$ at long wavelengths and flatter at short ones, likely due to the lack of turbulence at small scales and the influence of gravity waves at large scales. Nevertheless, the estimates of $\varepsilon$ can be found in Table3 for all flights and all layers.

### 3.4.2  Estimates from the velocity structure functions

An alternative, theoretically equivalent, way to estimate $\varepsilon$ comes from the analysis of the n-th order structure functions of velocity fluctuations:

$$S_n(l) = \langle |u(x+l) - u(x)| \rangle^n, \tag{5}$$

where $l$ is the distance. According to theory (e.g., Frisch (1995)) estimate of $\varepsilon$ from the n-th order structure function can be obtained from:

$$S_n(l) = C_n |l\varepsilon|^{n/3} \tag{6}$$

where $C_n$ is constant of the order of 1.

According to Kolmogorov theory for 3rd order structure function (n=3) constant $C_3 = 1$ and estimate of $\varepsilon$ does not need any empirical information, whereas for the 2nd-order structure function a knowledge of the actual value of constant $C_2$ is required. This constant is of the order of 1, but is different for longitudinal and transversal fluctuations. Chamecki and Dias (2004) give the appropriate values of $C_2t \approx 2$ for transverse velocity fluctuations and $C_2l \approx 2.6$ for longitudinal velocity fluctuations.

In practice, estimating from the 2nd-order is common for airborne measurements because the quality of the data is not sufficient to unambiguously determine scaling of the 3rd-order structure function. This was also the case in our data. We calculated the 2nd-order structure function for each layer and flight composite and used a linear fit with a slope of $2/3$ in the range of scales corresponding to the same range of frequencies as in estimates from PSD. Having variable directional wind shear at the cloud top, it was difficult find an unambiguous reference frame to define longitudinal and transverse fluctuations. We decided to use velocity fluctuations in the $x$ (East-West), $y$ (North-South) and $w$ (vertical) directions. Thus, only vertical fluctuations can be considered traversal, whereas both the u and v components contain a significant amount of longitudinal velocity fluctuations. Consequently, we used $C_2l$ for the horizontal fluctuations and $C_2t$ for the vertical ones, keeping in mind that the estimates we produce from these components can somewhat inaccurate. The second-order composite structure functions and suitable

fits for all flights, layers and velocity components are presented in Figure 6. The estimated by this method values of $\varepsilon$ complement Table3.

Estimates of $\varepsilon$ are plotted in Fig7 to facilitate the comparison across the cloud top layers, methods, velocity components and flights.

Generally, $\varepsilon$ estimates from the 2nd-order structure functions are less variable than those from the power spectra. The $\varepsilon$ profiles across the cloud top layers are overall consistent and in agreement with the distribution of TKE and squared velocity fluctuations: no dissipation in the FT, moderate dissipation in the TISL, typically maximum dissipation in the CTMSL and slightly smaller values in the CTL.

Signs of anisotropy (smaller variances in the vertical velocity fluctuations than in the horizontal ones) are clearly visible in the TISL and weakly noticeable in the CTMSL. Anisotropy is also reflected in the scaling ranges, larger for horizontal velocity fluctuations than for vertical ones. Interestingly, most of the 2nd-order structure function exhibit scale break around 100m, which confirms earlier assumption of a typical size of large eddies.

The values of $\varepsilon$ across the layers are large, often exceeding $10^{-3} m^2 s^{-3}$. This has important consequences, as discussed below.

## 4  Discussion

As documented by the analysis of 8 research flights from POST, with flight patterns containing many successive ascents and descents across the stratocumulus top region, the upper part of the STBL has a complex vertical structure. Algorithmic layer division based on experimental evidence (Malinowski et al., 2013) allowed the layers characterized by different thermodynamic and turbulent properties to be distinguished. The cloud top is separated from the free troposphere by the EIL, which consists of two sublayers. The first sublayer is the TISL, which is typically $\sim 20m$ thick (c.f. Tab.1), has strong inversion, is hydrostatically stable, yet turbulent. The source of turbulence in this layer is wind shear, spanning across the layer and reaching deeper into the cloud top. The bulk Richardson number across this layer in all investigated cases is close to the critical value. The layer is marginally unstable, suggesting that the thickness of the layer adapts to velocity and temperature differences between the uppermost part of the cloud and free troposphere. The turbulence in this layer is anisotropic, with vertical fluctuations damped by static stability and horizontal fluctuations enhanced by shear (c.f. Table4). The TKE dissipation rate $\varepsilon$ in the TISL is substantial, with typical values $\varepsilon \sim 2*10^{-4} m^2/s^3$. The TISL is void of clouds, i.e., it can be described with dry thermodynamics, as no evaporation occurs there. To interact with clouds, free tropospheric air must be transported by turbulence across the TISL, mixing with more humid air from just above the cloud top on the way.

Below the TISL, there is a CTMSL cohabitated by cloud top bubbles and volumes without cloud droplets (c.f. Figs. 3-7 in Malinowski et al. (2013)). The CTMSL is also hydrostatically stable on average, but the stability is weaker than that of the TISL. This layer is also affected by wind shear. As in the TISL, the bulk Richardson number across the layer is close to critical, i.e., less static stability is accompanied by less shear. Turbulence in this layer is also anisotropic, with reduced vertical fluctuations. Analyses of both the TKE itself and $\varepsilon$ indicate that the CTMSL is the most turbulent layer of the STBL top region. Cloud bubbles do not mix with free tropospheric air, but with cloud-free air preconditioned and humidified during turbulent transport across the TISL. Temperature and humidity differences between CTL and FT do not result in predicted buoyancy reversal due to preconditioning in FT, as indicated in recent analysis by Gerber et al. (2015). However, the thickness of CTMSL is somehow dependent on thermodynamic conditions in FT. The three thinnest CTMSLs were observed in flights where mixing of FT and CTL air could theoretically produce negative buoyancy (CTEI permitting conditions) - refer to Table 1 here and Table 4 in Gerber et al. (2013)). In contrast, in all other investigated cases, CTMSL is $\sim 2$ times thicker ($\sim 60m$ vs. $\sim 30m$).

As expected, turbulence is negligible in the FT and is strongly turbulent in the CTL. Turbulence in the CTL is isotropic. Porpoises with slightly positive Ri values indicate the production of turbulence by buoyancy.

### 4.1 Corrsin and Ozmidov scales

In the following, we focus on the TISL and CTMSL to better understand the effects of anisotropy. Following Smyth and Moum (2000), who analyzed turbulence in stable layers in the ocean, we estimate two turbulent length scales associated with stable stratification and shear. The first one, the Corrsin scale, is a scale above which turbulent eddies are deformed by the mean wind shear and is expressed as

$$L_C = \sqrt{\varepsilon/S^3}. \tag{7}$$

Here, $S$ is the mean velocity shear across the layer. The second one, the Ozmidov scale, is a scale above which eddies are deformed by stable stratification and is expressed as

$$L_O = \sqrt{\varepsilon/N^3}, \tag{8}$$

where $N$ is the mean Brunt-Vaisala frequency across the layer. The ratio of the Ozmidov and Corrsin scales is closely related to the Richardson number and can be estimated as follows, independent of $\varepsilon$:

$$\frac{L_C}{L_O} = \left(\frac{N}{S}\right)^{\frac{3}{2}} = Ri^{\frac{3}{4}}. \tag{9}$$

Histograms of these scales for all suitable porpoises and all flights, obtained with the estimated values of $\varepsilon$ for all three velocity components, are shown in Fig.8. The estimates of $N$, $S$, $\varepsilon$, $L_C$ and $L_O$ for all sublayers and flights are reported in Table 4. The most important finding is that the Ozmidov and Corrsin scales are smaller than $1m$ in the TISL. In fact, they are as small as $30cm$. This means that eddies of characteristic sizes above 30 cm are deformed by buoyancy and shear, which first act to reduce the eddies' vertical size and then expand the eddies in the horizontal direction. Turbulent eddies spanning the entire thickness of the TISL, i.e., $\sim 20m$ (if they exist), are significantly elongated in the horizontal direction. They do not transport mass across the layer effectively, and the existing temperature and humidity gradients indicate that the layer is not well mixed. We suspect that failures in the estimates of entrainment velocities in the STBL (as discussed in Wood (2012)), can be explained by the fact that few studies have focused on turbulence in the TISL. We hypothesize that mixing across this layer depends on the poorly understood dynamics of stably stratified turbulence (e.g., Rorai et al. (2014, 2015)). Thus, entrainment parameterizations should be revisited with this fact in mind. Whether the thermodynamic effects of the FT and CTL air result in buoyancy reversal is of secondary importance to mass flux and scalar fluxes across the TISL.

### 4.2 Buoyancy and shear Reynolds numbers

In scales smaller than $L_C$ and $L_O$ turbulence is not affected by anisotropy. The range of scales of isotropic turbulence spans down to Kolmogorov microscale $\eta$. "Its value can be estimated from the known TKE dissipation rate and air kinematic viscosity $\nu = 1.4607 * 10^{-5}[m^2/s]$ via:

$$\eta = \left(\frac{\nu^3}{\varepsilon}\right)^{1/4}. \tag{10}$$

Knowing the Kolmogorov microscale allows the characterization of small-scale turbulence in TISL and CTMSL by means of buoyancy and shear Reynolds numbers, $Re_B$ and $Re_S$ (for details consult e.g. Chung and Matheou (2012)) from the following formulas:

$$Re_B = \left(\frac{L_O}{\eta}\right)^{4/3} \tag{11}$$

$$Re_S = \left(\frac{L_C}{\eta}\right)^{4/3} \tag{12}$$

Estimates of $\eta$, $Re_B$, $Re_S$ are presented in the last columns of Tab.4. Clearly, range of scales of isotropic turbulence in CTMSL is much larger than that in TISL. As a rule of thumb it can be stated Kolmogorov microscale in CTMSL is as

small as $1.5mm$ and twice as large in TISL. Corresponding buoyancy and shear Reynolds numbers are of the order of $10^3$ in TISL and of the order of $3 * 10^4$ in CTMSL. In terms of Reynolds numbers and range of scales, small-scale turbulence in CTMSL is much more developed than that in TISL.

Finally, data collected in Tab.4 give some hints, potentially useful for improvements of entrainment/mixing parametrizations. Both $N$ and $S$ are in TISL roughly twice as large as in CTMSL. Thus, knowing the temperature and buoyancy jumps across the EIL the thickness of these layers can be estimated on a basis of critical Ri. Successful parametrization should include these parameters, which govern turbulence in the sublayers of the EIL and account for moisture jump, in order to account for thermodynamic effects of entrainment. It is disputable to which extent radiative cooling should be added, since its effects are most likely accounted for in the temperature jump. High resolution LES and/or DNS modelling of EIL turbulence should help in finding a functional form of an improved parametrization.

## 5    Conclusions

Using high-resolution data from cloud top penetrations collected during the POST campaign, we analyzed 8 different cases and investigated the turbulence structure in the vicinity of the top of the STBL. Using algorithmic layer division based on records of temperature, LWC and the three components of wind velocities, we found that the EIL, separating the cloud top from the free atmosphere, consists of two distinct sublayers: the TISL and the CTMSL. We estimated the typical thicknesses of these layers and found that the TISL was in the range of $15 - 35m$ and the CTMSL was in the range of $25 - 75m$. In both layers, turbulence is produced locally by shear and persists despite the stable stratification. The bulk Richardson number across the layers is close to critical, which confirms earlier hypotheses that the thickness of these layers adapts to large-scale forcings (by shear and temperature differences across the STBL top) to keep these layers marginally unstable in a dynamical sense. Additionally, the thickness of the CTMSL was found to be dependent on the humidity of FT. Both shear and stable stratification make turbulence in both layers highly anisotropic. Quantitatively, this anisotropy is estimated using the Corrsin and Ozmidov scales, and we found that these scales were as small as $\sim 30cm$ in the TISL and $\sim 3m$ in the CTMSL. Such small numbers clearly show that turbulence governing the entrainment of free tropospheric air is stably stratified and highly anisotropic on scales comparable to the layer thickness. In scales smaller than Corrsin and Ozmidov ones buoyant and shear Reynolds numbers indicate that turbulence in CTMSL is much more developed than that in TISL. An accurate description of the exchange between the STBL and FT requires a better understanding of the turbulence in both layers which

is significantly different with different sources and characteristics than that in the STBL below the cloud top region.

*Acknowledgements.* The POST field project was supported by the National Science Foundation through grant ATM-0735121 and by the Polish Ministry of Science and Higher Education through grant 186/W-POST/2008/0. The analyses of POST data presented here were supported by the Polish National Science Centre through grant DEC-2013/08/A/ST10/00291.

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

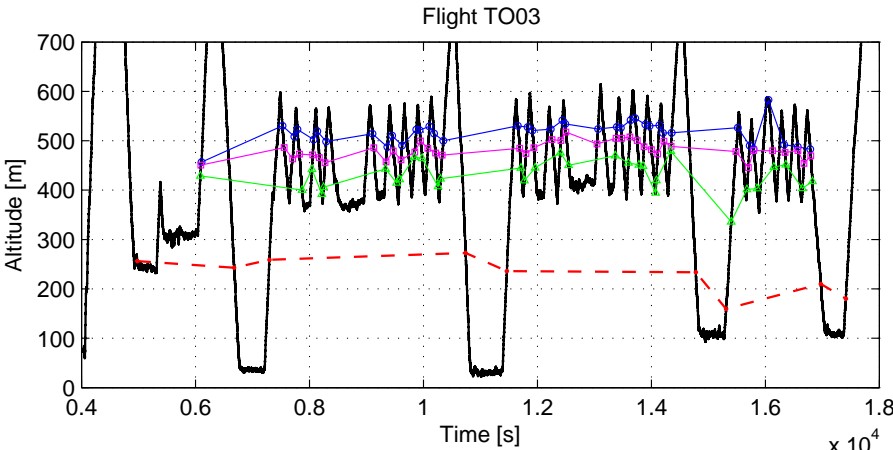

**Figure 1.** Vertical profiles of TO03 flight with the layer division superimposed. Blue marks indicate FT-TISL division on the porpoises, purple: TISL-CTMSL division, green: CTMSL-CTL division. All data points where the layer division algorithm gave unambiguous results are shown. The red dashed line indicates the cloud base.

**Table 1.** Flight info, layer division and thickness of the EIL sublayers estimated from cloud top penetrations. Flight - flight number; Type - brief information of the case type (N/N - Non-classical, Night, C/D - Classical, Day etc.); No porpoises - total number of porpoises through the cloud top in the area of the experiment; $\Delta T$ - temperature jump across the EIL; $\Delta q$ - humidity jump across the EIL, b - buoyancy of saturated mixture of cloud top and FT air; No TISL - number of successful detection of TISL on porpoises; TISL - thicknes of TISL, No CTMSL - number of successful detection of CTMSL on porpoises; CTMSL - thicknes of CTMSL. Thermodynamic parameters taken from Gerber et al. (2013).

| Flight | Type | No porpoises | $\Delta T$ [K] | $\Delta q$ [$g/kg$] | b [$ms^{-2}$] | No TISL | TISL [m] | No CTMSL | CTMSL [m] |
|--------|------|-------------|---------|-----------|----------|---------|-----------|----------|-----------|
| TO03 | N/N | 50 | 10.1 | -3.65 | 0.0048 | 39 | $35.1 \pm 18.0$ | 31 | $48.5 \pm 26.4$ |
| TO05 | N/N | 49 | 2.8 | -0.71 | 0.0161 | 27 | $16.7 \pm 22.5$ | 25 | $69.8 \pm 40.0$ |
| TO06 | C/N | 70 | 7.5 | -5.94 | -0.0059 | 58 | $13.9 \pm 7.4$ | 46 | $32.7 \pm 26.1$ |
| TO07 | N/D | 64 | 2.9 | -0.27 | 0.0171 | 22 | $19.6 \pm 16.3$ | 17 | $49.1 \pm 25.9$ |
| TO10 | C/D | 55 | 8.7 | -5.70 | -0.0033 | 53 | $25.0 \pm 10.5$ | 49 | $24.8 \pm 20.8$ |
| TO12 | C/N | 58 | 8.9 | -4.67 | -0.0001 | 42 | $23.1 \pm 9.9$ | 45 | $34.7 \pm 25.8$ |
| TO13 | N/N | 58 | 2.3 | -0.49 | 0.0175 | 31 | $14.3 \pm 14.3$ | 27 | $74.2 \pm 35.5$ |
| TO14 | N/N | 57 | 6.4 | -1.47 | 0.0123 | 37 | $22.0 \pm 10.7$ | 43 | $48.6 \pm 27.5$ |

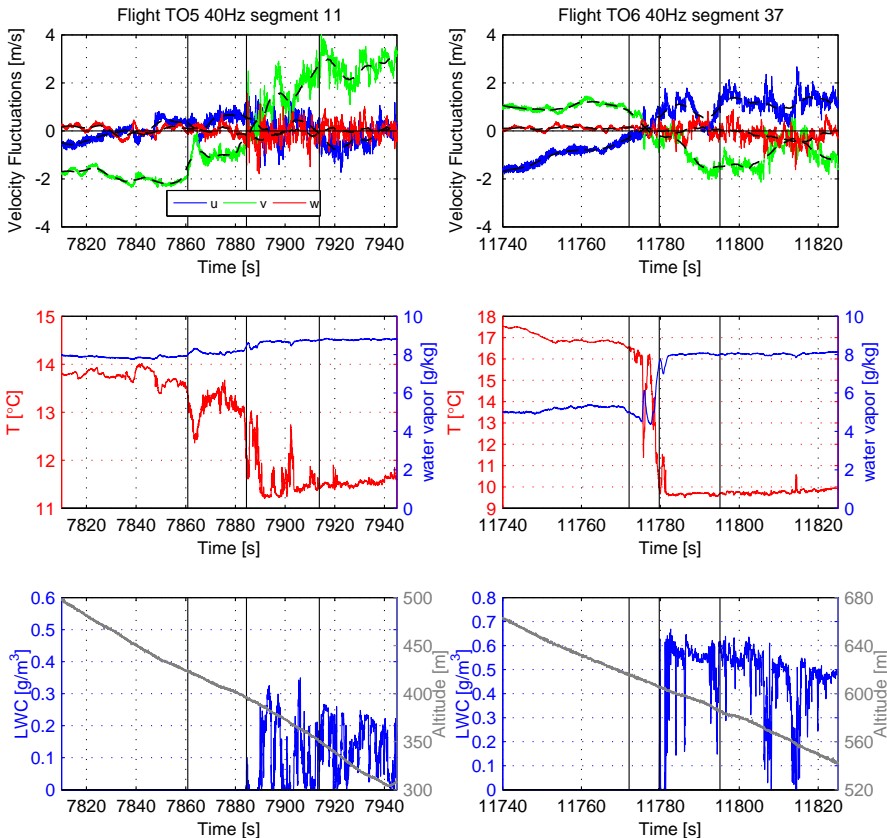

**Figure 2.** Layer division on example penetrations from TO05 ("non-classical") and TO12 ("classical") flights are shown in two columns. In top panels three components of wind velocity u,v,w recorded at a sampling rate of 40 Hz are presented in blue, green and red. Thick dashed lines represent centered running averages over 300 data points, black vertical lines resulting from the algorithmic layer division, layers (from the left): free troposphere (FT), Turbulent Inversion Sublayer (TISL), Cloud Top Mixing Sublayer (CTMSL), Cloud Top Layer (CTL).

In the middle panels corresponding temperature and humidity records are shown. In the lowest panel liquid water content and aircraft altitude are shown.

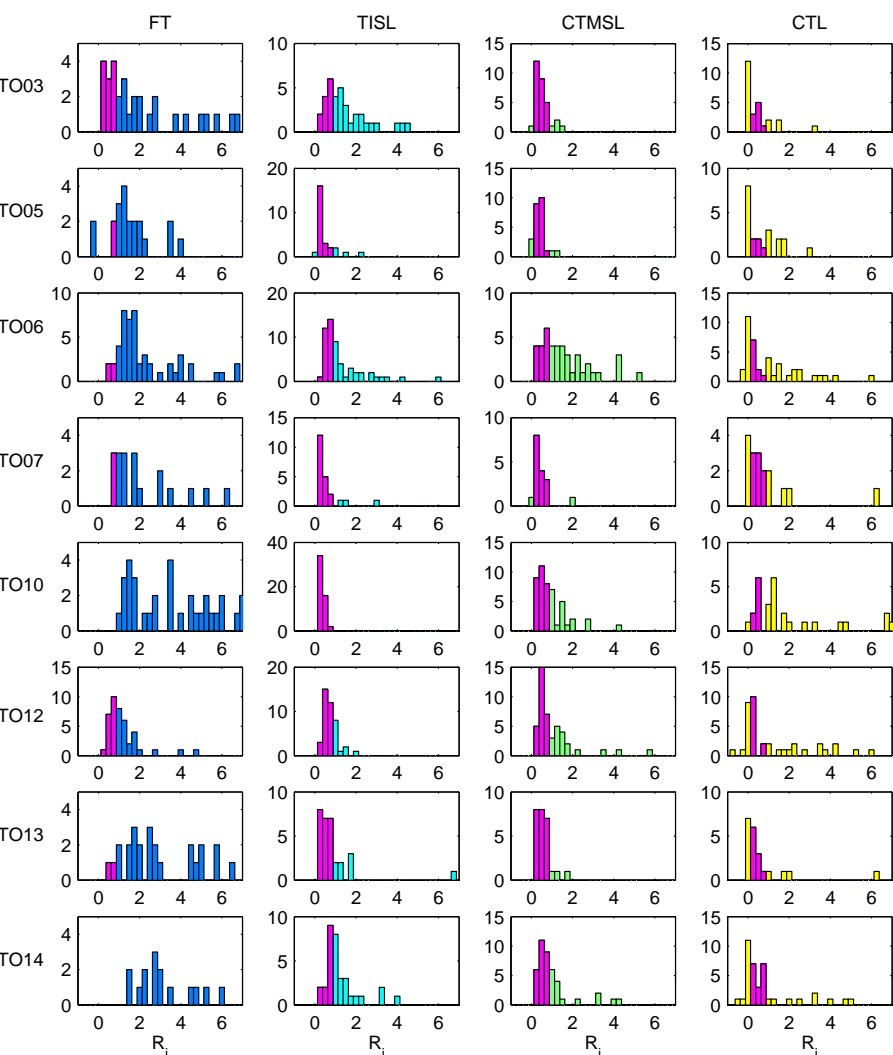

**Figure 3.** Histograms of the bulk Richardson numbers $R_i$ across the layers and sublayers of the stratocumulus top regions. Bins of $R_i$ centered at 0.25, 0.5 and 0.75, i.e., close to the critical value, are shown in magenta.

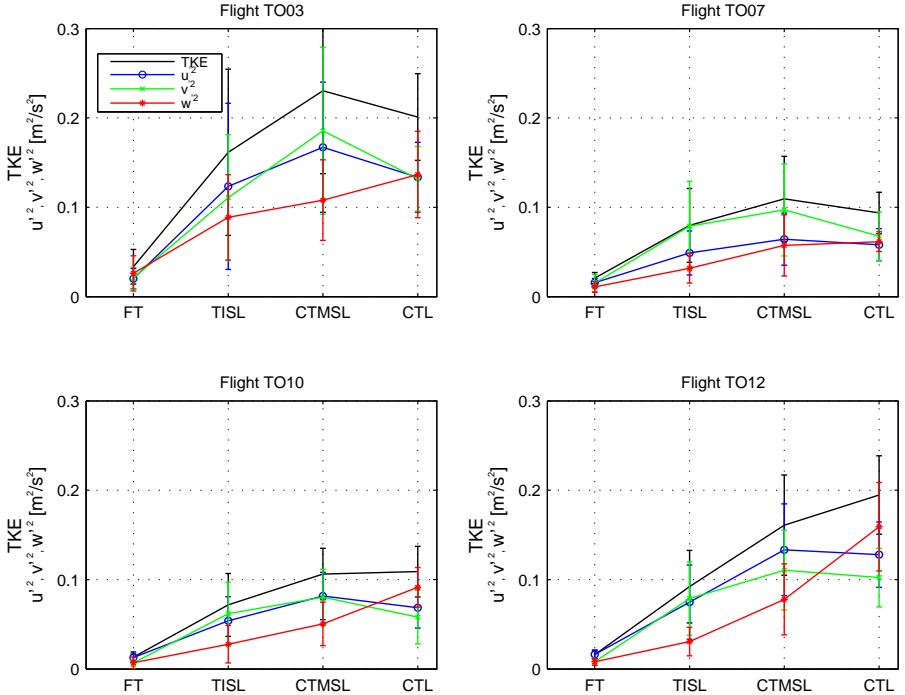

**Figure 4.** Four examples of turbulent kinetic energy (TKE) and squared average velocity fluctuations in consecutive sublayers of the STBL are presented. u,v,w, (blue, green, red) denote WE, NS and vertical velocity fluctuations, respectively.

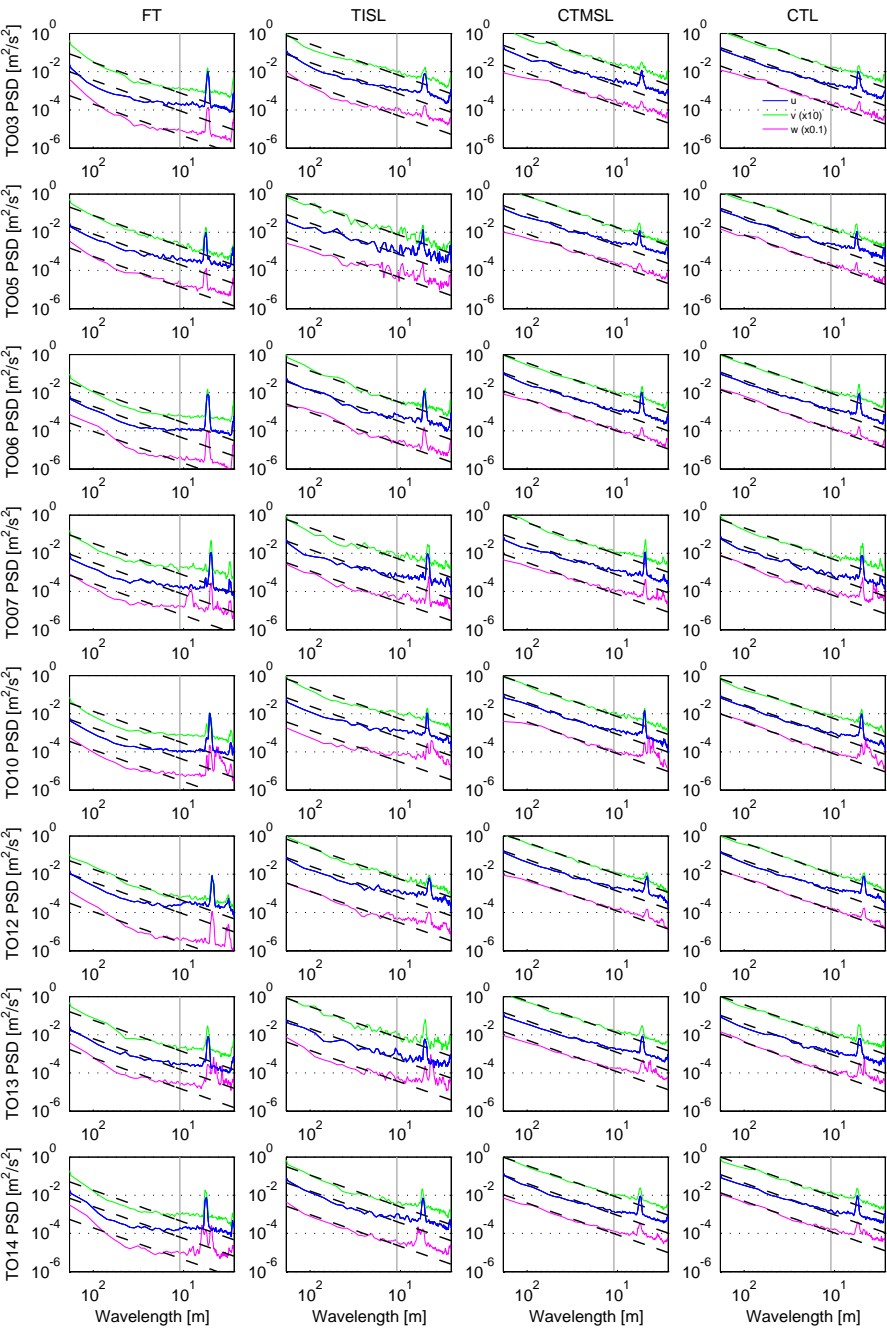

**Figure 5.** Power spectral density of the velocity fluctuations of the three components u, v, w, (blue, green, red) composites for all ascents/descents. Individual spectra are shifted by factors of 10 for comparison. Dashed lines show the -5/3 slope fitted to the spectra in a range of frequencies from 0.3 Hz to 5 Hz to avoid instrumental artefacts at higher frequencies.

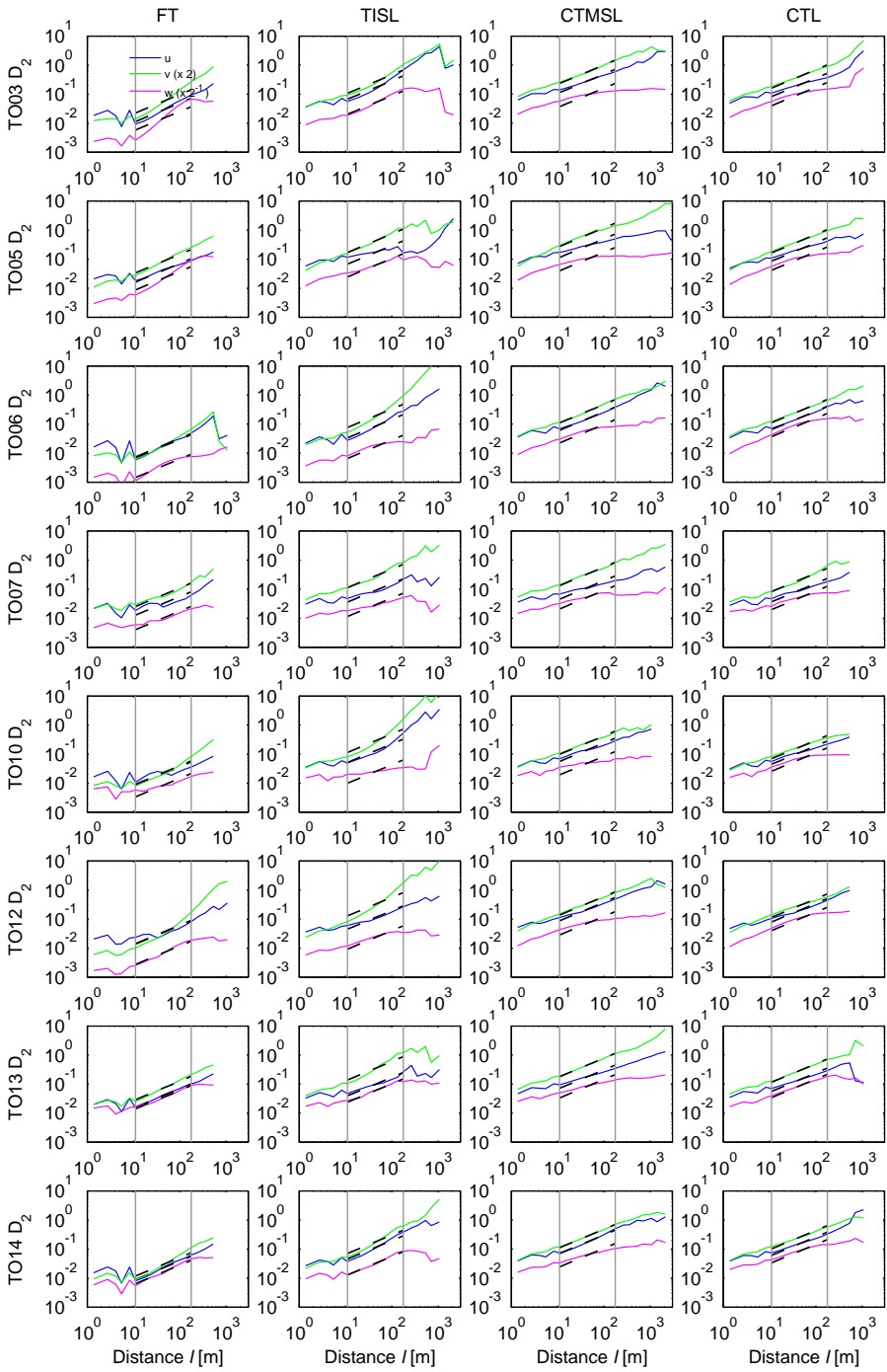

**Figure 6.** 2nd-order structure functions of the velocity fluctuations of three components u, v, w, (blue, green, red) composites for all ascents/descents. Individual structure functions are shifted by factors of 2 for comparison. Dashed lines show the 2/3 slope fitted to the functions in a range of frequencies from 0.3 Hz to 5 Hz (corresponding range of scales indicated by vertical solid lines) to avoid instrumental artefacts at higher frequencies.

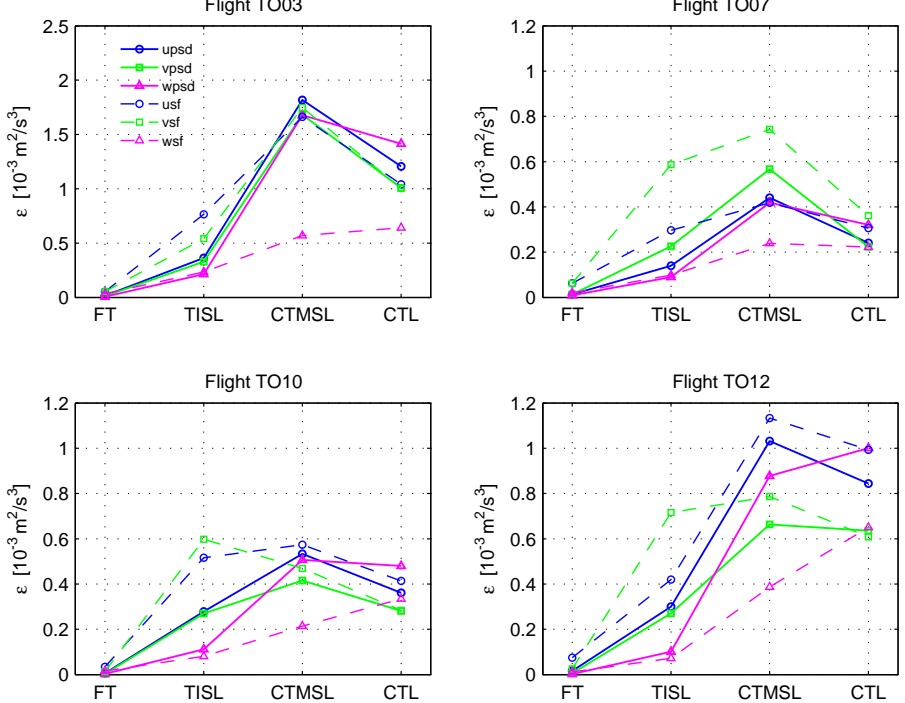

**Figure 7.** Example the estimates of the TKE dissipation rate $\varepsilon$ in sublayers for 4 selected flights. Continuous lines denote estimates based on the power spectral density, dashed lines indicate estimates from 2nd-order structure functions, and circles, squares and triangles indicate u,v and w velocity fluctuations, respectively.

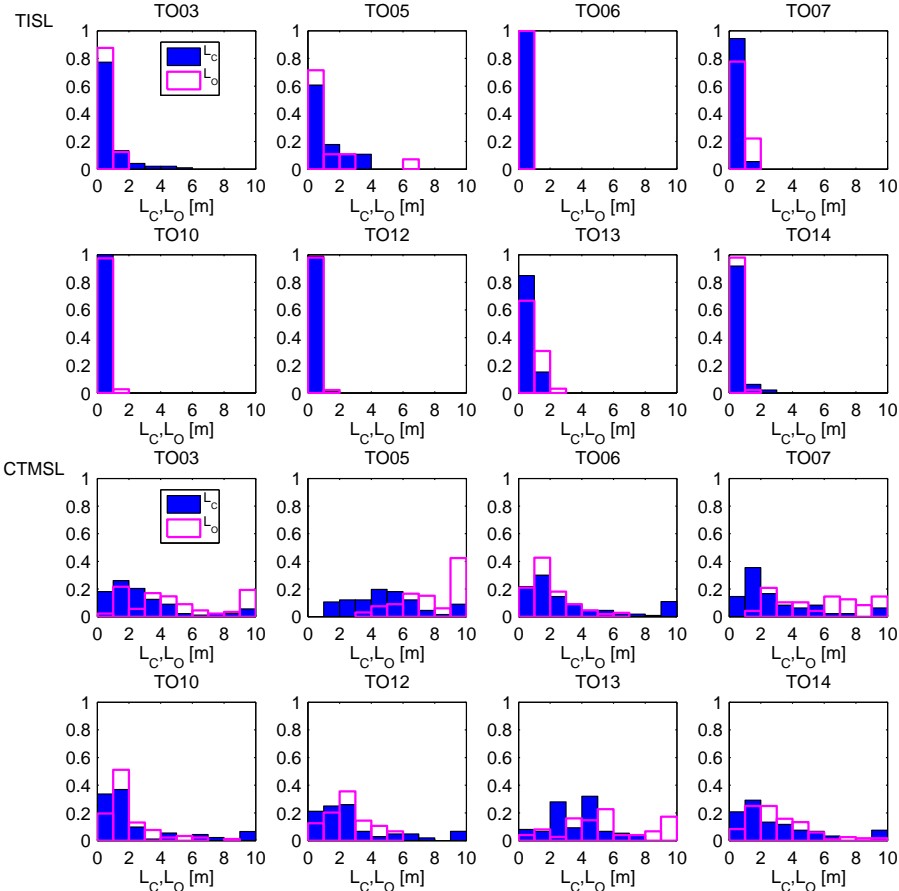

**Figure 8.** Histograms of the Corrsin (blue bars) and Ozmidov (empty red bars) scales in the TISL and CTMSL on porpoises for all investigated flights. Bins every 1 m.

**Table 2.** Root-mean-square fluctuations of the velocity components (u, v, w) and turbulent kinetic energy for different layers of the cloud top in all investigated POST flights, as defined in the text.

| Flights | Layers | u_RMS [m/s] | v_RMS [m/s] | w_RMS [m/s] | TKE [m2/s2] |
|---------|--------|-------------|-------------|-------------|-------------|
| TO03 | FT | $0.137 \pm 0.036$ | $0.139 \pm 0.040$ | $0.152 \pm 0.055$ | $0.033 \pm 0.019$ |
|  | TISL | $0.326 \pm 0.126$ | $0.306 \pm 0.106$ | $0.280 \pm 0.086$ | $0.161 \pm 0.093$ |
|  | CTMSL | $0.401 \pm 0.087$ | $0.420 \pm 0.108$ | $0.322 \pm 0.071$ | $0.230 \pm 0.093$ |
|  | CTL | $0.358 \pm 0.054$ | $0.362 \pm 0.053$ | $0.363 \pm 0.068$ | $0.201 \pm 0.049$ |
| TO05 | FT | $0.142 \pm 0.030$ | $0.137 \pm 0.066$ | $0.150 \pm 0.072$ | $0.038 \pm 0.035$ |
|  | TISL | $0.295 \pm 0.133$ | $0.356 \pm 0.182$ | $0.272 \pm 0.140$ | $0.195 \pm 0.146$ |
|  | CTMSL | $0.417 \pm 0.105$ | $0.486 \pm 0.146$ | $0.334 \pm 0.069$ | $0.266 \pm 0.133$ |
|  | CTL | $0.341 \pm 0.058$ | $0.348 \pm 0.073$ | $0.342 \pm 0.061$ | $0.183 \pm 0.056$ |
| TO06 | FT | $0.107 \pm 0.021$ | $0.077 \pm 0.021$ | $0.063 \pm 0.016$ | $0.012 \pm 0.005$ |
|  | TISL | $0.224 \pm 0.073$ | $0.216 \pm 0.073$ | $0.137 \pm 0.050$ | $0.068 \pm 0.032$ |
|  | CTMSL | $0.322 \pm 0.086$ | $0.313 \pm 0.079$ | $0.244 \pm 0.066$ | $0.133 \pm 0.035$ |
|  | CTL | $0.319 \pm 0.061$ | $0.309 \pm 0.047$ | $0.366 \pm 0.059$ | $0.169 \pm 0.042$ |
| TO07 | FT | $0.121 \pm 0.021$ | $0.118 \pm 0.035$ | $0.099 \pm 0.025$ | $0.021 \pm 0.006$ |
|  | TISL | $0.210 \pm 0.065$ | $0.259 \pm 0.104$ | $0.171 \pm 0.060$ | $0.080 \pm 0.041$ |
|  | CTMSL | $0.249 \pm 0.057$ | $0.306 \pm 0.087$ | $0.236 \pm 0.080$ | $0.109 \pm 0.048$ |
|  | CTL | $0.240 \pm 0.036$ | $0.255 \pm 0.051$ | $0.250 \pm 0.026$ | $0.094 \pm 0.023$ |
| TO10 | FT | $0.110 \pm 0.019$ | $0.076 \pm 0.020$ | $0.077 \pm 0.030$ | $0.013 \pm 0.006$ |
|  | TISL | $0.222 \pm 0.053$ | $0.235 \pm 0.068$ | $0.158 \pm 0.054$ | $0.072 \pm 0.035$ |
|  | CTMSL | $0.293 \pm 0.076$ | $0.293 \pm 0.099$ | $0.217 \pm 0.058$ | $0.106 \pm 0.029$ |
|  | CTL | $0.258 \pm 0.039$ | $0.235 \pm 0.050$ | $0.300 \pm 0.036$ | $0.109 \pm 0.028$ |
| TO12 | FT | $0.124 \pm 0.017$ | $0.082 \pm 0.021$ | $0.086 \pm 0.020$ | $0.016 \pm 0.005$ |
|  | TISL | $0.254 \pm 0.067$ | $0.261 \pm 0.076$ | $0.166 \pm 0.046$ | $0.092 \pm 0.041$ |
|  | CTMSL | $0.365 \pm 0.080$ | $0.339 \pm 0.089$ | $0.272 \pm 0.073$ | $0.161 \pm 0.056$ |
|  | CTL | $0.354 \pm 0.052$ | $0.313 \pm 0.050$ | $0.393 \pm 0.064$ | $0.195 \pm 0.044$ |
| TO13 | FT | $0.149 \pm 0.043$ | $0.142 \pm 0.048$ | $0.188 \pm 0.086$ | $0.046 \pm 0.043$ |
|  | TISL | $0.244 \pm 0.055$ | $0.293 \pm 0.121$ | $0.303 \pm 0.123$ | $0.134 \pm 0.073$ |
|  | CTMSL | $0.330 \pm 0.054$ | $0.389 \pm 0.092$ | $0.313 \pm 0.052$ | $0.184 \pm 0.056$ |
|  | CTL | $0.298 \pm 0.046$ | $0.314 \pm 0.053$ | $0.335 \pm 0.086$ | $0.157 \pm 0.045$ |
| TO14 | FT | $0.117 \pm 0.026$ | $0.095 \pm 0.027$ | $0.120 \pm 0.054$ | $0.021 \pm 0.011$ |
|  | TISL | $0.278 \pm 0.108$ | $0.244 \pm 0.099$ | $0.210 \pm 0.090$ | $0.102 \pm 0.057$ |
|  | CTMSL | $0.339 \pm 0.101$ | $0.300 \pm 0.060$ | $0.274 \pm 0.061$ | $0.148 \pm 0.050$ |
|  | CTL | $0.318 \pm 0.059$ | $0.301 \pm 0.056$ | $0.343 \pm 0.066$ | $0.159 \pm 0.050$ |

**Table 3.** TKE dissipation rate $[10^{-3}\frac{m^2}{s^3}]$ estimated from the energy spectra and 2nd- order structure functions of velocity fluctuations.

| Flight | Method | FT | | | TISL | | | CTMSL | | | CTL | | | EIL | | |
|---|---|---|---|---|---|---|---|---|---|---|---|---|---|---|---|---|
| | | u | v | w | u | v | w | u | v | w | u | v | w | u | v | w |
| TO3 | PSD | 0.01 | 0.01 | 0.01 | 0.36 | 0.33 | 0.21 | 1.82 | 1.68 | 1.68 | 1.21 | 1.01 | 1.41 | 1.10 | 0.98 | 0.84 |
| | SF2 | 0.05 | 0.05 | 0.04 | 0.77 | 0.54 | 0.23 | 1.66 | 1.75 | 0.57 | 1.04 | 1.00 | 0.64 | 1.25 | 1.07 | 0.40 |
| TO5 | PSD | 0.05 | 0.05 | 0.03 | 0.37 | 0.38 | 0.19 | 1.95 | 1.63 | 1.67 | 1.17 | 0.92 | 1.40 | 1.82 | 1.53 | 1.46 |
| | SF2 | 0.09 | 0.10 | 0.07 | 0.76 | 1.09 | 0.31 | 1.71 | 2.21 | 0.64 | 1.09 | 1.03 | 0.68 | 1.43 | 1.95 | 0.54 |
| TO6 | PSD | 0.01 | 0.003 | 0.002 | 0.11 | 0.12 | 0.06 | 0.54 | 0.47 | 0.66 | 0.62 | 0.51 | 0.82 | 0.42 | 0.37 | 0.36 |
| | SF2 | 0.02 | 0.01 | 0.004 | 0.27 | 0.33 | 0.04 | 0.66 | 0.56 | 0.27 | 0.72 | 0.58 | 0.57 | 0.52 | 0.50 | 0.17 |
| TO7 | PSD | 0.01 | 0.01 | 0.01 | 0.14 | 0.23 | 0.09 | 0.44 | 0.57 | 0.42 | 0.24 | 0.22 | 0.32 | 0.39 | 0.61 | 0.44 |
| | SF2 | 0.06 | 0.06 | 0.02 | 0.30 | 0.59 | 0.10 | 0.42 | 0.74 | 0.24 | 0.31 | 0.36 | 0.22 | 0.40 | 0.65 | 0.19 |
| TO10 | PSD | 0.01 | 0.003 | 0.003 | 0.28 | 0.27 | 0.11 | 0.53 | 0.42 | 0.51 | 0.36 | 0.28 | 0.48 | 0.41 | 0.38 | 0.25 |
| | SF2 | 0.03 | 0.01 | 0.02 | 0.52 | 0.60 | 0.08 | 0.57 | 0.47 | 0.21 | 0.41 | 0.28 | 0.33 | 0.58 | 0.60 | 0.14 |
| TO12 | PSD | 0.02 | 0.01 | 0.003 | 0.30 | 0.27 | 0.10 | 1.03 | 0.66 | 0.88 | 0.84 | 0.64 | 1.00 | 0.77 | 0.58 | 0.52 |
| | SF2 | 0.07 | 0.03 | 0.01 | 0.42 | 0.72 | 0.07 | 1.13 | 0.79 | 0.39 | 0.99 | 0.61 | 0.65 | 0.88 | 0.86 | 0.26 |
| TO13 | PSD | 0.03 | 0.03 | 0.03 | 0.22 | 0.36 | 0.13 | 0.89 | 0.97 | 0.86 | 0.53 | 0.53 | 0.59 | 0.82 | 0.96 | 0.75 |
| | SF2 | 0.09 | 0.08 | 0.13 | 0.35 | 0.80 | 0.29 | 0.84 | 1.18 | 0.49 | 0.58 | 0.61 | 0.51 | 0.72 | 1.14 | 0.46 |
| TO14 | PSD | 0.01 | 0.01 | 0.01 | 0.15 | 0.08 | 0.07 | 0.59 | 0.48 | 0.55 | 0.64 | 0.50 | 0.77 | 0.48 | 0.37 | 0.40 |
| | SF2 | 0.04 | 0.02 | 0.04 | 0.42 | 0.29 | 0.12 | 0.83 | 0.57 | 0.31 | 0.65 | 0.50 | 0.49 | 0.67 | 0.47 | 0.26 |

**Table 4.** Buoyancy, shear, TKE dissipation rates, Corrsin, Ozmidov and Kolmogorov scales and buoyancy and shear Reynolds numbers in TISL and CLMSL sublayers of the EIL. All symbols as in the text, No - number of penetrations on which estimates were obtained.

| Flight | Layer | No | $N[s^{-1}]$ | $S[s^{-1}]]$ | $\varepsilon[m^2s^{-3}*10^{-3}]$ | $L_C[m]$ | $L_O[m]$ | $\eta[mm]$ | $Re_B$ | $Re_S$ |
|---|---|---|---|---|---|---|---|---|---|---|
| TO03 | TISL | 34 | 0.09±0.02 | 0.09±0.07 | 0.30±0.39 | 0.89±0.96 | 0.55±0.37 | 2.39±1.01 | 2600 | 4600 |
| | CTMSL | 29 | 0.04±0.02 | 0.07±0.04 | 1.46±1.49 | 3.03±2.63 | 5.16±3.37 | 1.33±0.25 | 78000 | 39000 |
| TO05 | TISL | 9 | 0.05±0.02 | 0.13±0.07 | 0.27±0.69 | 1.04±1.08 | 1.29±1.51 | 2.67±0.87 | 11000 | 4100 |
| | CTMSL | 22 | 0.03±0.01 | 0.06±0.05 | 1.70±1.49 | 5.34±3.32 | 9.25±3.87 | 1.24±0.18 | 160000 | 82000 |
| TO06 | TISL | 35 | 0.11±0.01 | 0.11±0.04 | 0.07±0.12 | 0.25±0.21 | 0.21±0.18 | 3.32±1.02 | 500 | 530 |
| | CTMSL | 36 | 0.06±0.02 | 0.06±0.04 | 0.43±0.24 | 3.54±4.25 | 1.98±1.31 | 1.74±0.34 | 14000 | 36000 |
| TO07 | TISL | 13 | 0.06±0.02 | 0.10±0.05 | 0.12±0.13 | 0.41±0.24 | 0.75±0.40 | 2.79±0.85 | 2600 | 1070 |
| | CTMSL | 16 | 0.02±0.01 | 0.05±0.02 | 0.46±0.40 | 3.07±2.66 | 6.14±3.62 | 1.78±0.35 | 68000 | 28000 |
| TO10 | TISL | 41 | 0.10±0.01 | 0.17±0.04 | 0.18±0.23 | 0.18±0.13 | 0.38±0.26 | 2.53±0.79 | 1300 | 480 |
| | CTMSL | 32 | 0.06±0.02 | 0.08±0.04 | 0.38±0.20 | 2.59±3.43 | 1.90±1.42 | 1.77±0.25 | 14000 | 24000 |
| TO12 | TISL | 30 | 0.10±0.01 | 0.13±0.03 | 0.16±0.25 | 0.30±0.21 | 0.35±0.23 | 2.67±0.87 | 1200 | 900 |
| | CTMSL | 35 | 0.05±0.02 | 0.07±0.04 | 0.75±0.43 | 3.13±3.21 | 2.58±1.27 | 1.51±0.25 | 24000 | 36000 |
| TO13 | TISL | 10 | 0.07±0.02 | 0.11±0.06 | 0.32±0.92 | 0.59±0.45 | 0.73±0.56 | 2.64±0.83 | 3900 | 2300 |
| | CTMSL | 25 | 0.03±0.02 | 0.05±0.02 | 0.85±0.45 | 3.60±1.72 | 5.64±2.86 | 1.46±0.24 | 72000 | 39000 |
| TO14 | TISL | 33 | 0.09±0.01 | 0.09±0.04 | 0.09±0.16 | 0.45±0.44 | 0.31±0.24 | 3.06±0.83 | 800 | 1400 |
| | CTMSL | 41 | 0.04±0.01 | 0.05±0.03 | 0.47±0.24 | 3.63±4.91 | 3.07±1.89 | 1.68±0.24 | 27000 | 43000 |