# Peer review of "Physics of Stratocumulus Top (POST): turbulence characteristics"

_Atmospheric Chemistry and Physics, 2015_

## Referee Comment (RC1) · Anonymous Referee #1 · 3 Feb 2016

This study analyzes turbulence properties of the EIL by decomposing it into two sublayers based on the POST observation data. Their analysis confirms existence of shear generated turbulence in the EIL, and suggests adjustment of the EIL so that the bulk Richardson number is maintained near critical value. Also, the authors show anisotropic turbulence in the EIL due to damped vertical fluctuations by static stability. While their analysis is valid, two of these main results are not new, so I think that the authors should perform further analysis so that this study is considered to be published in ACP. For instance, why the algorithm does not successfully divide the EIL into two sublayers for all cases, but only 8 cases? How is the assumption for the characteristic horizontal size of large eddies of the order of approximately 100 m justified? Why the classical cases show long tails in the CTMSL (figure 3). TO14 also has longer tail. Why the theoretically equivalent method to estimate the TKE dissipation rate gives

sometimes very different results? What is a better way to incorporate their findings into entrainment parameterization? Another concern is that, although I see some usefulness to study these two sublayers, I am not fully convinced if decomposing the EIL into two sublayers is absolutely necessary, since their main results seem to hold for the bulk of the EIL. In other words, their motivation to study two sublayers is rather weak and the significance of analyzing these two layers is not fully appreciated. This criticism partly comes from the lack of discussion for Tables 2, 3, and 4 and Fig. 6. Size of figures are too small. Showing many plots in one figure is not always a good way.

Specific comments

Correct "turbulent kinetic energy" to "turbulence kinetic energy.

line 45: "...that stratocumulus clouds often persist..." has been reported since Kuo and Schubert (1988) so this is not "recent."

line 57: "wind shear in and above the cloud top is another important ..." is partly comes from the updrafts diverging below the inversion layer, just mentioned previous sentence. At least, remove "another."

line 98: "highly turbulent" but in the abstract, "marginally turbulent." Marginal is not high.

line 150: Is "this model" the classical cases?

line 159: Are TO10 and TO13 "extreme" cases? Are they "well representing" classical or non-classical cases?

line 254: Briefly describe "the EIL structure."

line 314: Define "RMS."

line 341: Reference for "numerical simulations of the TO13..."?

line 349: What is the motivation to estimate the TKE dissipation rate?

line 380: "The second and third..." is unclear.

line 387: "Spectra..." is unclear

line 415: Add "Sn(l) = Cn |l epsilon|ˆ(n/3). Also define l, x, n, Cn.

line 425&426": Are "(4x18/55)" for Ct & "(4/3*4*24/55)" for Cl necessary? Without explanation, they are meaningless.

line 471: "substantially" turbulent so what? generate enough mixing?

Fig. 1: Show only one case with larger size.

Fig. 2: Convert time to height if possible. Add potential temperature, water vapor, and cloud water profiles. Make panels larger, they are too small.

Fig. 3: Add histogram for EIL. Select a few cases (e.g., 2 for non-classical, 2 for classical). Make plots larger, they are too small.

Fig. 4: Panels are too small. In stead of plotting u'2 and v'2, plot horizontal component of TKE, i.e., (u'2+v'2)/2, unless there are notable differences between u'2 and v'2. Also plot the EIL values.

Fig. 5: Select and show one case. The figure is too small. Correct the legend since it overlaps the curves. Also plot the EIL's PSD.

Fig. 6: Fix as suggested for Fig. 5

Fig. 7: Make the figure larger. In stead of plotting the TKE dissipation for u and v, plot the dissipation rate for the horizontal component of TKE. Also, plot the EIL values.

Fig. 8: The figure size is too small. Select a few cases.

Table 1: Add three columns for EIL depth, classical or not, and CTEI or not.

Table 2: Add a row for each flight for EIL values. In stead of listing u and v, can they be combined? These values are similar.

Table 3: List the dissipation rate for the horizontal component of TKE, in stead of u and v component unless notable differences between u and v component exists.

Table 4: Add EIL values.

[Figure]

---

## Author Comment (AC1) · 8 Feb 2016

We thank the anonymous referee for his/her review of our work. We will improve the manuscript according to most of the referee's suggestions, which we find inspiring, important and valuable. While we respectfully disagree with the general comment that there is not enough material in the paper to justify the publication, we agree that the additional analyses and discussion suggested shall improve the paper's quality.

**Referee1, general comment:**

*"This study analyzes turbulence properties of the EIL by decomposing it into two sublayers based on the POST observation data. Their analysis confirms existence of shear generated turbulence in the EIL, and suggests adjustment of the EIL so that the bulk Richardson number is maintained near critical value. Also, the authors show*

[Figure]

*anisotropic turbulence in the EIL due to damped vertical fluctuations by static stability. While their analysis is valid, two of these main results are not new, so I think that the authors should perform further analysis so that this study is considered to be published in ACP."*

**Our answer:**

We agree with the referee that the present paper extends earlier study, but we believe that there are more new findings other than the ones noticed by the reviewer. From our point of view:

1) We extended former preliminary results concerning shear generated turbulence in stably stratified inversion, layer division and bulk Richardson number Ri based on two cases only to a wider range of stratocumulus top conditions, making these findings not hypothetical but more robust and well documented.

2) We provide important new information on the thicknesses of cloud top sublayers.

3) Most importantly, we characterize properties of turbulence in the sublayers by numbers, providing so far unknown information on anisotropy of turbulence, TKE and its components and estimates of TKE dissipation rates. We show that turbulence characterized by these numbers is DIFFERENT in sublayers, despite the fact that thicknesses of turbulent inversion sublayer (TISL) and cloud top mixing sublayer (CTMSL) result from near critical value of Ri.

4) We characterize anisotropy of turbulence in cloud top sublayers by means of Ozmidov and Corrsin scales, showing that these scales reach minimum of few tens of centimeters in TISL and of single meters in CTMSL. Such characterization of Sc top has not been documented so far.

We do not resist further analysis, but we disagree with the opinion that only a minority of the paper contains new findings.

**Questions and suggestions:**

1) *"...why the algorithm does not successfully divide the EIL into two sublayers for all cases, but only 8 cases?"*

This is a misunderstanding, and we will add explanations in the revised text. Our algorithm works well in all the cases we investigated. We limited ourselves to 8 cases due to practical reasons: workload to perform the analysis is enormous and resources are limited. Thus, from all flights we selected 8 cases covering the whole spectrum of physical conditions observed during the experiment.

2) *"How is the assumption for the characteristic horizontal size of large eddies of the order of approximately 100 m justified?"*

Some justification comes from the small influence of the averaging length on final results, which is written in the text. More justification can be found in power spectral densities and structure functions of vertical velocity fluctuation in TISL and CTMSL showing signatures of weak scale break at about 100 m. We will discuss this point in the revised text.

3) *"Why the classical cases show long tails in the CTMSL (figure 3). TO14 also has longer tail"*

This is most likely related to the accuracy of the estimate of the shear across thin layers. We will analyze this and discuss it in the revised text.

4) *"Why the theoretically equivalent method to estimate the TKE dissipation rate gives sometimes very different results?"*

In our opinion methods used to estimate the TKE dissipation rate are theoretically equivalent only in homogeneous, isotropic, stationary and neutrally stratified turbulence, which is not the case in our study. In the manuscript we write: *"Derivation of the TKE dissipation rate from moderate-resolution airborne measurements is always problematic. The assumptions of isotropy, homogeneity and stationarity of turbulence, used to calculate the mean TKE dissipation rate from power spectra and/or structure*

[Figure]

*functions, are hardy, if ever, fulfilled. This is also the case in our investigation of highly variable thin sublayers of the STBL top and is enhanced by the porpoising flight pattern. Considering these problems, we estimated the TKE dissipation rate by two methods. Three spatial components of velocity fluctuations are treated separately, allowing for the study of possible anisotropy, which is expected due to the different stability and shear in the stratocumulus top sublayers."* Nevertheless we will add additional discussion.

5) *"What is a better way to incorporate ... findings into entrainment parameterization?"*

This is a complex question, worthy of a new paper when answered. At this moment we may only state that we see a need to perform very high resolution (close to Corrsin and Ozmidov scales) numerical simulations of the cloud top region in order to understand how eddies that are anisotropic by shear and static stability transport mass, and how exactly the exchange between Sc top and free atmosphere looks. We will elaborate on this in the revised text.

6) *"Another concern is that, although I see some usefulness to study these two sublayers, I am not fully convinced if decomposing the EIL into two sublayers is absolutely necessary, since their main results seem to hold for the bulk of the EIL. In other words, their motivation to study two sublayers is rather weak and the significance of analyzing these two layers is not fully appreciated. This criticism partly comes from the lack of discussion for Tables 2, 3, and 4 and Fig. 6."*

We will provide more discussion to better document the importance of division into the sublayers.

We thank the reviewer for the specific comments and will account for them in the revised manuscript.

---

## Referee Comment (RC2) · Anonymous Referee #2 · 26 Feb 2016

Summary:

This paper analyses turbulence characteristics near the stratocumulus cloud top from aircraft observations in the POST campaign. The authors previous study developed a method to decompose the entrainment interfacial layer (EIL) into two sublayers, and tested it on two POST research flights. The present paper extends this analysis to six further research flights. Whilst this increases the robustness of the previous results, I am unsure whether that fact alone makes this study worthy of publication. Therefore I feel the authors either need to do some further analysis, or a better job of highlighting what exactly is novel about the current paper, before it can be considered suitable for publication. I have added some thoughts / ideas below.

Specific points:

The previous study (Malinowski et al 2013) considered two contrasting profiles as examples of possible stratocumulus states. I don't see the justification for choosing the additional six that were used here. How were these flights chosen? Were they the ones the method worked best for? If so it would be useful to document the flights where the method didn't work and reasons for this. Are these two sublayers universal features of stratocumulus cloud tops, or only present under certain circumstances? Why not use all POST flights, to give a much larger sample size and allow a more statistical analysis of the results?

It would be interesting to show on Table 1 the total number of cloud top penetrations in that flight, to see how frequently the method is diagnosing these layers. What happens on T007, where it looks like you diagnose layers on less than one-third of the cloud top penetrations? It would also be interesting to have some discussion of the difference between numbers in TISL and CTMSL diagnoses, i.e. what is happening when one is found but not the other?

One of the clearest reasons (to me) for considering these two sublayers came from the difference in the Corrsin and Ozmidov scales in the two sub-layers, yet very little is made of this result and could perhaps be expanded upon. What does the much larger, and more varied, lengthscales in the CTMSL tell you about that region of the cloud top?

All the plots could be bigger and clearer.

Minor/typo:

L31 - should say "aims" instead of "aimed"

L33 - I'd suggest removing "the"

L88 - I'd suggest defining "porpoising" the first time you use it, for readers who may be unfamiliar with the term

L91 - Gerber et al shouldn't be in brackets

L125 - Perhaps say where Monterey Bay is for readers who may not know

L146 - should say "cloud" instead of "clod"

L177 - I'd suggest adding a comma after "wind shear"

L199 - should be 17-58 cloud top penetrations

L456/482 - the 10ˆpower is a bit messed up

L554 - should say "mind" instead of "find"

---

## Referee Comment (RC3) · Anonymous Referee #3 · 3 Mar 2016

The manuscript discusses turbulence characteristics in the region of the cloud-top of stratocumulus clouds. The in-situ measurements from the POST campaign are used. The analysis is organized by considering the layer division classification of Malinowski et al. (2013), who carried out a closely related analysis of the POST dataset. The paper has two main goals/contributions: (a) derive small-scale and high-order turbulence statistics, such as the kinetic energy dissipation and velocity structure functions and (b) Using the POST dataset analysis, inform approaches for understanding and modeling the cloud-top entrainment process.

The study of stratocumulus clouds is of significant interest because of their large radiative forcing and the paper should be of interest to the broader atmospheric science community. The effort to process and analyze the in-situ measurements is also significant, because, as the authors point out, these measurements are difficult and valuable.

[Figure]

Overall, I find the paper interesting and a valuable addition to the stratocumulus literature. However, the paper has a significant limitation: it is mostly a presentation of processed results from the POST campaign. Most of the conclusions are expected and the investigation does not have significant depth in terms of links to theory. I believe that the current manuscript is not suitable for publication in Atmospheric Chemistry and Physics. A substantially revised version of the manuscript can be suitable for publication.

Major comments:

1. The analysis of the results is presented without any reference or relation to the broader meteorological conditions. The results, such as the dissipation rates of Figure 7, show large variability between flights. The authors seem to suggest that the bulk Richardson number and a second parameter based on neutrally stratified dynamics (they use the Corrsin scale) are sufficient to characterize the data. If this is the suggestion, it should by made clearer and explicit.

2. Further to the previous point, there is no information about the broader large-scale environment. For instance: the authors report zonal and meridional velocity statistics but there are meaningless without a reference direction. These should be presented with respect to the direction of shear.

3. Some information about the nature of convection and the radiative forcing of the cloud top should be included to make the presentation more self contained.

4. One of the conclusions is that "Turbulence in both sublayers is highly anisotropic, with Corrsin and Ozmidov scales...". I think it is well-established that the largest turbulent motions in the inversion are anisotropic. In fact, this is what the Ozmidoc and Corrsin scales characterize: the smallest scale where the effects of stable stratification and shear are important. The interesting question is if there is enough separation of scales from L_O or L_C to the Kologorov scale for the turbulence to approach isotropy at small scales. This is important for modeling, because many turbulence closures assume small-scale isotropy and Kolmogorov scaling. It is perhaps beneficial to consider also the buoyancy Reynolds number (see eq. 1.3 and related discussion in Chung & Matheou, 2012, Journal of Fluid Mechanics), in addition to the length scales.

5. All the analysis is carried out under the assumption that stratification and shear are the dominant processes and that radiative cooling and buoyancy modification by latent heat exchange (e.g. buoyancy reversal) are neglected. This should be better justified. On line 60 these other processes are mentioned and it is argued that "These multiple sources are responsible for exchange across the inversion."

6. Further, assuming that radiative cooling and latent hear exchange does not play a significant role, why are the results not appropriately scaled? Most of the results are reported in dimensional quantities. Some of the scaling in Chung & Matheou (2012) and references therein can apply to the current data.

7. The definition of the Richardson number in eq. 1 should be based on the virtual potential temperature, rather than just the potential temperature.

Minor comments:

1. All the figures are very difficult to read.

2. On line 48: "Turbulent transport across the inversion is a mechanism that limits exchange between the cloud top and free atmosphere and should be considered". This sentence is not clear and it means that turbulent transport limits the transport of mass, momentum and energy across the inversion. Perhaps the intention is to say that stable stratification limits turbulent transport across the inversion.

3. On line 95: "These measurements indicated that wind shear across the EIL is a source of turbulence" Wind shear is a source of turbulence, the measurements are not needed to show this.

4. On line 100: I think a better term is "an empirically based division" rather than "an experimentally based division"

5. On line 165: perhaps is better to use the term "aircraft"

6. In Figure 1, in the caption, I am not sure what "The corresponding lines indicate segment-averaged" means. Which lines are the authors referring to?

7. In figure 6 the logarithm of distance and structure function are used as the x-y axis, rather than plots with logarithmic x-y scale. Currently, the x-axis has units of logarithm distance, which is strange.

8. In Table 3 and the corresponding text that refers to the table, it is not very clear what u, v and w mean.
* * *

---

## Author Comment (AC2) · 7 Mar 2016

We thank the reviewer for his/her comments. While we will revise the manuscript accounting for his/her suggestions in order to improve the discussion of our results, we cannot agree that the paper is a straightforward extension of the previous one. The previous paper did not provide estimates of turbulent kinetic energy and velocity components variances, TKE dissipation rate and detailed characterization of turbulence anisotropy across cloud top layers. This experimental characterization, obtained on rich statistics of cloud top penetrations in various conditions, documents variability of turbulence from cloud top to free troposphere above in a way never, to our knowledge, done before.

**Specific comments:**

[Figure]

**1)** *"Therefore I feel the authors either need to do some further analysis, or a better job of highlighting what exactly is novel about the current paper, before it can be considered suitable for publication."* -

In order to highlight the new findings mentioned above we will expand the abstract, the introduction and conclusions to make them stand out to the reader.

**2)** *"The previous study (Malinowski et al 2013) considered two contrasting profiles as examples of possible stratocumulus states. I don't see the justification for choosing the additional six that were used here. How were these flights chosen? Were they the ones the method worked best for? If so it would be useful to document the flights where the method didn't work and reasons for this. Are these two sublayers universal features of stratocumulus cloud tops, or only present under certain circumstances? Why not use all POST flights, to give a much larger sample size and allow a more statistical analysis of the results?"*

We already partly answered these questions in our reply to the reviewer 1. After a laborious processing before undertaking the analysis, we selected data for the analysis from all POST flights to cover the whole span of key cloud top parameters: temperature and humidity jumps, wind shear and buoyancy effects of mixing. Despite the fact that we were not always able to distinguish between the sublayers (as briefly explained below) we will discuss flights selection process in the revised text.

**3)** *"It would be interesting to show on Table 1 the total number of cloud top penetrations in that flight, to see how frequently the method is diagnosing these layers. What happens on T007, where it looks like you diagnose layers on less than one-third of the cloud top penetrations? It would also be interesting to have some discussion of the difference between numbers in TISL and CTMSL diagnoses, i.e. what is happening when one is found but not the other?"*

We will add the required info to Tab. 7. Frankly speaking, there are several possible reasons of failure, not related to the method principle: too shallow porpoises (either

too low the uppermost point or too high the lowermost point) and small inclination of aircraft trajectory i.e. effects of horizontal inhomogeneities from e.g. superposition of boundary layer turbulence/circulation. After the layer division we performed additional quality control so that only unambiguous cases are included into the analysis. The problem with the discussion is that we have no comparison, i.e. we cannot say which porpoises were too shallow and how much, and there is no way to distinguish between too shallow porpoises and horizontal variability. We will add this information into the text.

**4)** *"One of the clearest reasons (to me) for considering these two sublayers came from the difference in the Corrsin and Ozmidov scales in the two sub-layers, yet very little is made of this result and could perhaps be expanded upon. What does the much larger, and more varied, length scales in the CTMSL tell you about that region of the cloud top?"*

The reasons for considering layer division were identified on a basis of temperature, humidity, wind and LWC time series in porpoises and discussed in Malinowski 2013. Physical reasons include different shear and static stability across the layers, different amplitudes temperature fluctuations and last but not least effects of cloud presence: dry mixing in TISL vs. moist mixing with possible evaporative cooling in CTMSL as well as radiative cooling in CTMSL. In this paper we show that application of such layer division makes additional sense and allows characterization of differences in turbulence properties within the layers, such as velocity variances and TKE, dissipation rate and finally Corrsin and Ozmidov scales. We did not elaborate a lot on Corrsin and Ozmidov scales, since we wanted to provide experimental results, not speculations. We found the contents of the paper already sufficiently inclusive. We presently perform high-resolution numerical simulations we performed to better understand the meaning of these findings. Nevertheless, we will add more discussion on differences between the sublayers and suggest possible consequences.

---

## Author Response (AR1)

Dear Editor, dear Reviewers,

in the attached documents please find our response to the reviewer comments.

We thank the referees for their reviews of our work. We substantially modified the manuscript according to the majority of the suggestions, which we find inspiring, important and valuable. We believe that the additional analyses and discussions added improved the paper's quality, hoping that the paper in the present shape it is worth publication in ACP.

In order to facilitate evaluation of the changes in the manuscript we also attach to this reply DIFF file, where changes in the manuscript are clearly marked: removed parts are in red, new parts are in blue. Suitable parts of DIFF file are also copy-pasted in order to make our reactions to the comments clearly exposed. We attached also a revised manuscript. Notice, please, that line numbers in the DIFF file and revised manuscript do not agree. In the answers to the reviewers we use numbering from the DIFF file.

Sincerely

Szymon Malinowski

**Reply to the Referee 1**

We thank the Referee for the in-depth review. Below there is a detailed description of our actions undertaken to modify the manuscript along the reviewer suggestions.

**General comments:**

1) This study analyzes turbulence properties of the EIL by decomposing it into two sublayes based on the POST observation data. Their analysis confirms existence of shear generated turbulence in the EIL, and suggests adjustment of the EIL so that the bulk Richardson number is maintained near critical value. Also, the authors show anisotropic turbulence in the EIL due to damped vertical fluctuations by static stability. While their analysis is valid, two of these main results are not new, so I think that the authors should perform further analysis so that this study is considered to be published in ACP.

We agree with the Referee that the present paper begins from the extension of the earlier study, but this is only the first step. A new results concerning layer thickness, TKE across Sc layers, TKE dissipation rate, Corrsin and Ozmidov scales are now better underlined. We added also additional analyses of Kolmogorov scales and Reynolds numbers across the layers (new section 4.2 and additional information in Table 4).We took the effort to better expose these new findings in the text. In particular we reworded and extended the abstract, c.f. lines 7-29 in the p.1 of the DIFFERENCE file:

estimated. The data are used to calculate turbulence characteristics, including the bulk Richardson number, meansquare velocity fluctuations, turbulent turbulence kinetic en-

- 10 ergy (TKE), and estimates of the TKE dissipation rate, and Corrsin, Ozmidov and Kolmogorov scales. A comparison of these properties among different sublayers indicates that the entrainment interfacial layer consists of two significantly different sublayers: the turbulent inversion sublayer (TISL) and
- the moist, yet statically hydrostatically stable, cloud top mixing sublayer (CTMSL). Both sublayers are marginally turbulent; , i.e. the bulk Richardson number across the layers is critical. This means that turbulence is produced by shear and damped by buoyancy such that the sublayer thicknesses
- 20 adapt to temperature and wind variations across them. Turbulence in both sublayers is highly anisotropic, with Corrsin and Ozmidov scales as small as  $\sim 30cm$  and  $\sim 3m$  in the TISL and CTMSL, respectively. These values are  $\sim 60$  and  $\sim 15$  times smaller than typical layer depths, indicating
- flattened large eddies and suggesting no direct mixing of cloud top and free tropospheric air. Also, small scales of turbulence are different in sublayers as indicated by the corresponding values of Kolmogorov scales and buoyant and shear Reynolds numbers.

We added also additional explanations in the introduction, consult p. 2, lines 65-77 of the DIFF file:

In the present paper, using algorithmic layer division, 65 we extend we begin from extension of the analysis of the POST data by Malinowski et al. (2013) to a larger number of cases. Then, we discuss performance of the algorithmic layer division, allowing for objective distinction of cloud top sublayers. As a main part of the study we analyze the proper-70 ties of turbulence in the sublayers to provide an experimental detailed characterization of turbulence in the stratocumulus cloud top region, based on a wide range of measurement data. Finally, we discuss the consequences of the fine structure of the turbulent cloud top and capping inversion, with a focus on 75 the vertical variability of turbulence and characteristic length scales.

**2) For instance, why the algorithm does not successfully divide the EIL into two sublayers for all cases, but only 8 cases?**

This is a misunderstanding. Our algorithm works well in all the cases we investigated. We limited ourselves to 8 cases due to practical reasons: workload to perform the analysis is enormous and resources are limited. Thus, from all flights we selected 8 cases covering the whole spectrum of physical conditions observed during the experiment. We added explanations concerning the data selection (p.3, I.28-49 of DIFF):

| from the Using Tables 1, 2 and 4 of Gerber et al. | (2013          | from          |
|---------------------------------------------------|----------------|---------------|
| all 17 POST flight we selected 8 cases (T         | '003, ' | TO05 , |

- TO06, TO07, TO10, TO12, TO13, TO14), which cover the whole range of observed temperature and humidity jumps across the inversion, shear strengths, cloud top change rates, entrainment velocities, buoyancies of cloud-clear air mixtures and day/night conditions (c.f. Tab] for
- key parameters). For these cases we repeated analyses of Malinowski et al. (2013) performing layer division, and estimating Richardson Numbers across the layers. Then, in order to understand dynamics of mixing process, we determined turbulence characteristics in the layers. We used
- 40 measurements of three components of wind velocity and fluctuations. These data were collected, sampled at a rate of 40 Hz using with a five-hole gust probe and corrected for the motion of the plane (Khelif et al. 1999). The features and differences of these characteristics among the
- 45 eloud top layersand flight case studies are discussed aircraft (Khelif et al.) [1999). We estimated values of Turbulence Kinetic Energy (TKE) and velocity variances in the layers, TKE dissipation rates, and finally, characterized anisotropy of turbulence.

We also added explanations why algorithm fails on some porpoises (p3, I.80-96 in DIFF):

Sometimes either division between FT and TISL or division 80 between CTMSL and CTL was not detected. This was most probably a result of too shallow individual porpoises. Before the experiment, in the course of discussion of flight pattern, it was decided that porpoises should be within a range of sim100 m from the cloud top. Actual 85 decision to stop ascent or descent was taken by the pilot based on this recommendation. A posteriori, in seems that sometimes slightly deeper porpoises would be more appropriate. Division algorithm, proposed on a basis of the available data, disregarded division points detected too close 90 to the local extremum of the aircraft altitude in order to avoid false estimates of the wind shear (division CTMSL/CTL) and TKE or temperature gradient (FT-TISL).

The example effect of the division are algorithm is plotted in Fig[1and, while all results, together with additional information about flights are summarized in Tab[1] In total,

3). How is the assumption for the characteristic horizontal size of large eddies of the order of approximately 100 m justified?

In p.7 I.2-6 of the diff file we added justification based on the results obtained:

Anisotropy is also reflected in the scaling ranges, larger for horizontal velocity fluctuations than for vertical ones. Interestingly, most of the 2nd-order structure function exhibit

 scale break around 100m, which confirms earlier assumption of a typical size of large eddies.

4) Why the classical cases show long tails in the CTMSL (figure 3). TO14 also has longer tail.

This is a problem of a thin layer, influencing estimates of gradients. Errors in the detection of the position of the shear layer, results in large effect due to the division by a small number, particularly important for shear which is in a power of 2 in denominator. We decided not to elaborate on this, since it was discussed in Malinowski et al., 2013.

**5) Why the theoretically equivalent method to estimate the TKE dissipation rate gives sometimes very different results?**

The methods used to estimate the TKE dissipation rate are theoretically equivalent only in homogeneous, isotropic, stationary and neutrally stratified turbulence, which is not the case in our study. In the manuscript we write: "Derivation of the TKE dissipation rate from moderate-resolution airborne measurements is always problematic. The assumptions of isotropy, homogeneity and stationarity of turbulence, used to calculate the mean TKE dissipation rate from power spectra and/or structure functions, are hardy, if ever, fulfilled. This is also the case in our investigation of highly variable thin sublayers of the STBL top and is enhanced by the porpoising

flight pattern. Considering these problems, we estimated the TKE dissipation rate by two methods. Three spatial components of velocity fluctuations are treated separately, allowing for the study of possible anisotropy, which is expected due to the different stability and shear in the stratocumulus top sublayers."

We introduced many small changes across the whole Sec. 3.4 to make these problems more clear, see, please DIFF file.

**6) What is a better way to incorporate their findings into entrainment parameterization?**

This is a complex question, worthy of a new paper when answered. We added some hints which might be useful in future studies in lines 43-55 of p.8 of DIFF file:

Finally, data collected in Tab 4 give some hints, potentially useful for improvements of entrainment/mixing parametrizations. Both N and S are in TISL roughly twice as large as in CTMSL. Thus, knowing the temperature 45 and buoyancy jumps across the EIL the thickness of these layers can be estimated on a basis of critical Ri. Successful parametrization should include these parameters, which govern turbulence in the sublayers of the EIL and account for moisture jump, in order to account for thermodynamic effects 50 of entrainment. It is disputable to which extent radiative cooling should be added, since its effects are most likely accounted for in the temperature jump. High resolution LES and/or DNS modelling of EIL turbulence should help in finding a functional form of an improved parametrization. 55

7) Another concern is that, although I see some usefulness to study these two sublayers, I am not fully convinced if decomposing the EIL into two sublayers is absolutely necessary, since their main results seem to hold for the bulk of the EIL. In other words, their motivation to study two sublayers is rather weak and the significance of analyzing these two layers is not fully appreciated. This criticism partly comes from the lack of discussion for Tables 2, 3, and 4 and Fig. 6.

We have already partially addressed this criticism answering to the comments above. In particular, we hope that the new section 4.3 is helpful, in particular I.32-41 in p.8:

Estimates of  $\eta$ ,  $Re_B$ ,  $Re_S$  are presented in the last columns of Tab4 Clearly, range of scales of isotropic turbulence in CTMSL is much larger than that in TISL. As a rule of thumb

- it can be stated Kolmogorov microscale in CTMSL is as small as 1.5mm and twice as large in TISL. Corresponding buoyancy and shear Reynolds numbers are of the order of  $10^3$  in TISL and of the order of  $3 * 10^4$  in CTMSL. In terms of Reynolds numbers and range of scales, small-scale
- turbulence in CTMSL is much more developed than that in TISL.

To underline better the importance of the EIL division into TISL and CTMSL we modified section 2.1 (connecting it with section 2.2) and added discussion (I. 104-106 in p.3 and I1-19 in p.4 in DIFF):

In order to illustrate the rationale for the layer division in Fig2 we present two randomly selected cloud penetrations from "non-classical" TO5 and "classical"

105

TO12 cases (another examples can be found in Malinowski et al. (2013)). Wind shear across the whole EIL present in both cases, usually weaker across CTMSL than across TISL. Wind velocity fluctuations in TISL are

- 5 less significant than in CTMSL. TISL is characterized by large mean temperature gradient (high static stability) and remarkable temperature fluctuations in dry environment. In CTMSL only a weak mean temperature gradient is present, temperature fluctuations are small, but the layer is moist
- 10 and LWC rapidly fluctuates between the maximum value for cloud and zero. Such striking differences indicate that division of the EIL into two sublayers is fully justified. But another question may arise: is division between CTMSL and CTL justified? The answer is yes, and the first part of
- the proof is in Malinowski et al. (2013), who show that turbulence in CTMSL is marginal in terms of Richardson number analysis. For more arguments behind this division let's investigate turbulence in both sublayers and adjacent FT and CTL.

**8) Size of figures are too small. Showing many plots in one figure Is not always a good way.**

We modified a majority of the figures to fulfill this requirement and the specific comments of the Referee. We decided to leave all panels in Fig.3, 5 and 6 since we think that they illustrate the variety of collected data and give information on the spread of the results and the details of each panel can be accurately seen by zooming the pdf file. Nevertheless we enlarged sizes of these figures to make them better visible on a printout as well.

**Specific recommendations.**

We accounted for almost all specific recommendations. The exception is no new plots and table columns with data for the whole EIL. In the revised version of the manuscript we present arguments and additional results indicating that sublayers of the EIL are very different. In such situation providing average data for the EIL could be misleading.

**Reply to the Referee 2**

We thank the reviewer for his/her comments. We revised the manuscript accounting for his/her suggestions in order to improve the discussion of our results. Below there is a detailed description of our actions undertaken to modify the manuscript along the reviewer suggestions.

**Specific comments:**

1) Therefore I feel the authors either need to do some further analysis, or a better job of highlighting what exactly is novel about the current paper, before it can be considered suitable for publication.

We extensively revised the manuscript, providing both: deeper description of the results and additional analyses. In particular we extended the abstract and the introduction and added additional analyses concerning layer division (substantial extensions of sections 1, 2, 2.1, and a new section 4.2). Since this requirement of the Referee 2 is similar to that of the Referee 1, we ask the reviewer to look for the detailed description of the changes into our reply to the Referee 1, remark 1.

2) The previous study (Malinowski et al 2013) considered two contrasting profiles as examples of possible stratocumulus states. I don't see the justification for choosing the additional six that were used here. How were these flights chosen? Were they the ones the method worked best for? If so it would be useful to document the flights where the method didn't work and reasons for this. Are these two sublayers universal features of stratocumulus cloud tops, or only present under certain circumstances? Why not use all POST flights, to give a much larger sample size and allow a more statistical analysis of the results?

We already answered these questions in our reply to the Referee 1. After a laborious processing before undertaking the analysis, we selected data for the analysis from all POST flights to cover the whole span of key cloud top parameters: temperature and humidity jumps, wind shear and buoyancy effects of mixing. For the details c.f. our answer to the Referee 1, remark 2.

3) It would be interesting to show on Table 1 the total number of cloud top penetrations in that flight, to see how frequently the method is diagnosing these layers. What happens on T007, where it looks like you diagnose layers on less than one-third of the cloud top penetrations? It would also be interesting to have some discussion of the difference between numbers in TISL and CTMSL diagnoses, i.e. what is happening when one is found but not the other?

We added the required info to Tab. 1, and, as mentioned above, added the discussion of flight selection and performance of the algorithm.

4) One of the clearest reasons (to me) for considering these two sublayers came from the difference in the Corrsin and Ozmidov scales in the two sub-layers, yet very little is made of this result and could perhaps be expanded upon. What does the much larger, and more varied, length scales in the CTMSL tell you about that region of the cloud top? Again, actions undertaken to satisfy this request ale already described in our answer to the Referee 1 (see our answer to remark 7). In particular: in the new section 2.1 we added a discussion on rationale of division EIL = TISL+CTMSL (I. 104-106 in p.3 and I1-19 in p.4 in DIFF) and according to the suggestion of the Referee 3 we added informations to Table 1 and Table 4 and wrote the new section 4.2 discussing results in the sublayers and possible recommendation for further studies aimed at better understanding of entrainment/mixing problematic.

**5) All the plots could be bigger and clearer.**

We modified a majority of the figures to fulfill this requirement as well as the specific comments of the Referee 1 We decided to leave all panels in Fig.3, 5 and 6 since we think that they illustrate the variety of collected data and give information on the spread of the results and the details of each panel can be accurately seen by zooming the pdf file. Nevertheless we enlarged sizes of these figures to make them better visible on a printout as well.

**Reply to the Referee 3**

We thank the Referee for the in-depth review, in particular for remarks 4 and 6. They allowed for an additional analysis, now included in Section 4.2. Below there is a detailed description of our actions undertaken to modify the manuscript along the reviewer suggestions.

**General comment:**

However, the paper has a significant limitation: it is mostly a presentation of processed results from the POST campaign. Most of the conclusions are expected and the investigation does not have significant depth in terms of links to theory.

We agree that most, or at least some conclusions were expected. Nevertheless expected does not mean documented. In our opinion the strength of the paper is not in links to the theory, but in the evidence based documentation of the expected effects. Taking all proportions, recent confirmation of gravity waves existence was expected, nevertheless documentation of their existence was an achievement.

Clearly, the above remark does not mean that we do not want to improve the manuscript and we seriously accounted for major comments, including those connected to better description of the results.

**Major comments:**

1. The analysis of the results is presented without any reference or relation to the broader meteorological conditions. The results, such as the dissipation rates of Figure 7, show large variability between flights. The authors seem to suggest that the bulk Richardson number and a second parameter based on neutrally stratified dynamics (they use the Corrsin scale) are sufficient to characterize the data. If this is the suggestion, it should by made clearer and explicit.

As you can see, we found that across TISL and CTMSL the Bulk Richardson number is critical, thus its value is not necessary to characterize the data. What is really necessary is temperature and wind jumps from CTL to FT, which allow characterization of the sublayers thickness, as well as TKE dissipation rate, which allows for the characterization of the scales: Corrsin, Ozmidov and finally Kolmogorov (thank you for the hint). Additional suggestion how shear and buoyancy are divided between the sublayers can be deduced from the improved Table 4. We used this deduction and wrote in the new Section 4.2 the following analysis (c.f. p.8, I.31-46 in DIFF file):

Estimates of  $\eta$ ,  $Re_B$ ,  $Re_S$  are presented in the last columns of Tab4 Clearly, range of scales of isotropic turbulence in CTMSL is much larger than that in TISL. As a rule of thumb

- it can be stated Kolmogorov microscale in CTMSL is as small as 1.5mm and twice as large in TISL. Corresponding buoyancy and shear Reynolds numbers are of the order of  $10^3$  in TISL and of the order of  $3 * 10^4$  in CTMSL. In terms of Reynolds numbers and range of scales, small-scale
- turbulence in CTMSL is much more developed than that in TISL.

Finally, data collected in Tab 4 give some hints, potentially useful for improvements of entrainment/mixing parametrizations. Both N and S are in TISL roughly twice as large as in CTMSL. Thus, knowing the temperature and buoyancy jumps across the EIL the thickness of these layers can be estimated on a basis of critical Ri. Successful

2. Further to the previous point, there is no information about the broader large-scale environment. For instance: the authors report zonal and meridional velocity statistics but there are meaningless without a reference direction. These should be presented with respect to the direction of shear and buoyancy jumps are divided between the sublayers can be deduced from the improved Table 4.

In order to account for this remark we added the following lines at the end of p.2 (DIFF):

(2012); Gerber et al. (2013). Meteorological conditions in the course of the measurements were stable in the Eastern North Pacific high pressure area with cloud tops were located between 375m and 760m (mean is  $513 \pm 137m$ ), stable wind direction (between 320 and 340 degrees) and speeds (6.5 - 14.5m/s) at the cloud top height, with the wind shear (sometimes directional) above cloud tops. Typical temperature at the cloud top was  $10.8^{\circ}C$ , temperature jumps across the inversion varied in a range 2.3 - 10.2K. More details concerning conditions in the course of flights can be

with the references to the processed and raw data in the beginng of p.3 (DIFF): found in tables 1-4 of Gerber et al. (2013) and in the open POST database (http://www.eol.ucar.edu/projects/post/).

We also modified text in p. 6 (DIFF):

transformed Having variable directional wind shear at the cloud top, it was difficult find an unambiguous reference frame to define longitudinal and transverse fluctuations. We decided to use velocity fluctuations in the x (East-West), y(North-South) and w (vertical) directions. Thus, only vertical fluctuations can be considered traversal, whereas both the u and v components contain a significant amount of longitudinal velocity fluctuations. ThusConsequently, we used  $C_t$  $C_2l$  for the horizontal fluctuations and  $C_t - C_2t$  for the vertical ones, keeping in mind that the estimates we produce from these components can somewhat inaccurate. The secondand added columns with temperature and humidity jumps to Table 1.

3. Some information about the nature of convection and the radiative forcing of the cloud top should be included to make the presentation more self contained.

To satisfy this requirement we added columns with character of the flight (Day/Night) and buoyancy of the saturated mixture of cloud top and FT air (possibility of buoyancy reversal due to mixing) is now included.

4. One of the conclusions is that "Turbulence in both sublayers is highly anisotropic, with Corrsin and Ozmidov scales. . .". I think it is well-established that the largest turbulent motions in the inversion are anisotropic. In fact, this is what the Ozmidoc and Corrsin scales characterize: the smallest scale where the effects of stable stratification and shear are important. The interesting question is if there is enough separation of scales from L\_O or L\_C to the Kologorov scale for the turbulence to approach isotropy at small scales. This is important for modeling, because many turbulence closures assume small-scale isotropy and Kolmogorov scaling. It is perhaps beneficial to consider also the buoyancy Reynolds number (see eq. 1.3 and related discussion in Chung & Matheou, 2012, Journal of Fluid Mechanics), in addition to the length scales.

Thank you for this remark. Accounting for it led to a new section 4.2 of the manuscript and following modification of conclusions. In particular, we discuss not only large scales (above Corrsin and Ozmidov ones) but also small scales between Lo/Lc and Kolmogorov. Performing this analysis throw more light on substantial difference between turbulence in TISL and CTMSL.

5. All the analysis is carried out under the assumption that stratification and shear are the dominant processes and that radiative cooling and buoyancy modification by latent heat exchange (e.g. buoyancy reversal) are neglected. This should be better justified. On line 60 these other processes are mentioned and it is argued that "These multiple sources are responsible for exchange across the inversion."

Well, this is not the assumption, but the result of bulk Richardson number analysis across the sublayers, especially across CTMSL. We were surprised when we first got this result (Malinowski et al., 2013, and suty of additional flights performed in this manuscript were aimed at better validation of this result, which we underlined in the new segment in p. 4:

for cloud and zero. Such striking differences indicate that division of the EIL into two sublayers is fully justified. But another question may arise: is division between CTMSL and CTL justified? The answer is yes, and the first part of

the proof is in Malinowski et al. (2013), who show that turbulence in CTMSL is marginal in terms of Richardson number analysis. For more arguments behind this division let's investigate turbulence in both sublayers and adjacent FT and CTL.

Later we deal with the problem of radiative cooling, writing in the section 4.2 the following (p.8, lines 51-53):

of entrainment. It is disputable to which extent radiative cooling should be added, since its effects are most likely accounted for in the temperature jump. High resolution LES

6. Further, assuming that radiative cooling and latent hear exchange does not play a significant role, why are the results not appropriately scaled? Most of the results are reported in dimensional quantities. Some of the scaling in Chung & Matheou (2012) and references therein can apply to the current data.

Thanks again for the suggestion, we added Section 4.2 and columns with Kolmogorov scales and Reynolds numbers to Table 4. we also changed the abstract to account for these new results.

7. The definition of the Richardson number in eq. 1 should be based on the virtual potential temperature, rather than just the potential temperature. Changed. In fact in calculations we were using virtual potential temperature.

**Minor comments:**

1. All the figures are very difficult to read.

Figures were improved, for the details see our reaction to the detailed comments of the Referee 1.

**Physics of Stratocumulus Top (POST): turbulence characteristics**

I. Jen-La Plante1, Y-F. Ma1, K. Nurowska1, H. Gerber2, D. Khelif3, K. Karpinska1, M. K. Kopec1, W. Kumala1, and S. P. Malinowski1

[revised manuscript text omitted]

5 Stationarity and horizontal homogeneity are accounted for when constructing the constructing composite PSDs for each layer by adding the summing individual PSDs for all suitable penetrations.

Each power spectrum from penetration

- through the investigated layer, P(f), is calculated using the Welch method in MATLAB with a moving window of  $2^8$  points on the 40 Hz velocity data. For This is done individually for each component of the velocity, the The fluctuations are determined with respect to a moving aver-
- 15 age of 300 points, as in the layer division. Spectra from all penetrations in a given layer and flight are Then each velocity spectrum fulfilling the quality criterion for each velocity component is combined into a composite spectrum , and then, for every flight. Finally the -5/3 line is fitted in
- 20 log-log coordinates. Figure 5 shows all the composite power spectra on a logarithmic scale, with the three velocity components spread out by factors of 10. The line with a slope -5/3 indicated by equation 4 is shown by the dashed line fits in the figure. The fit is limited to the frequency range
- of 0.3 5Hz, neglecting the higher frequency features attributed to interactions with the plane (and the lower frequency artifacts artefacts of the Welch method). The spectra in the CTMSL and CTL correspond well with the -5/3 law in the analyzed range of scales. A weak deviation - decreased
- amplitude small amplitude decrease of vertical velocity fluctuations at frequencies below 0.3 - 1Hz (depending on the flight) can be observed in the CTMSL. In the TISL, the scaling of velocity fluctuations with the -5/3 law is less evident; various deviations from a constant slope are more evi-
- dent in some flights (TO03, TO07, TO10, TO13) than in others. In the FT, scaling is poor; specifically, the spectra are steeper than -5/3 at long wavelengths and flatter at short ones, likely due to the lack of turbulence at small scales and the influence of gravity waves at large scales. Nevertheless,
- the estimates of  $\varepsilon$  can be found in Table3 for all flights and all layers.

**3.4.2 Estimates from the velocity structure functions**

An alternative, theoretically equivalent, way to estimate  $\varepsilon$  comes from the analysis of the n-th order structure functions of velocity fluctuations:

45

$$S_n(l) = \left\langle \left| u(x+l) - u(x) \right| \right\rangle^n, \tag{5}$$

where l is the distance. According to theory (e.g., Frisch (1995)) estimate of  $\varepsilon$  from the 3rd n-th order structure function can be obtained from:

$$S_{\underline{3n}}(l) = \underline{4/3l} \underbrace{C_n}_{\sim} \left| l\varepsilon \right|_{\sim}^{n/3} \tag{6}$$

where  $C_n$  is constant of the order of 1.

does not require any empirical constants, whereas the estimate from According to Kolmogorov theory for 3rd order structure function (n=3) constant  $C_3 = 1$  and estimate of  $\varepsilon$  does not need any empirical information, whereas for the 2nd-order structure function  $\tau$

$$S_2(l) = C_2 \left| \underline{l\varepsilon} \right|^{2/3}$$

requires a knowledge of the empirical actual value of constant  $C_2$ , which is on 
[revised manuscript text omitted]

|--------|---------------|--------------|----------------|-------------------|--------------|---------|---------------|----------------|---------------|
| TO03   | N/N           | 50           | 10.1           | -3.65             | 0.0048       | 39      | $35.1\pm18.0$ | 31             | $48.5\pm26.4$ |
| TO05   | N/N           | 49           | 2.8            | -0.71             | 0.0161       | 27      | $16.7\pm22.5$ | 25             | $69.8\pm40.0$ |
| TO06   | C/N           | 70           | 7.5            | -5.94             | -0.0059      | 58      | $13.9\pm7.4$  | 46             | $32.7\pm26.1$ |
| TO07   | N/D           | 64           | 2.9            | -0.27             | 0.0171       | 22      | $19.6\pm16.3$ | 17             | $49.1\pm25.9$ |
| TO10   | C/D           | 55           | 8.7            | -5.70             | -0.0033      | 53      | $25.0\pm10.5$ | 49             | $24.8\pm20.8$ |
| TO12   | C/N           | 58           | 8.9            | -4.67             | -0.0001      | 42      | $23.1\pm9.9$  | 45             | $34.7\pm25.8$ |
| TO13   | N/N           | 58           | 2.3            | -0.49             | 0.0175       | 31      | $14.3\pm14.3$ | 27             | $74.2\pm35.5$ |
| TO14   | N/N           | 57           | 6.4            | -1.47             | 0.0123       | 37      | $22.0\pm10.7$ | 43             | $48.6\pm27.5$ |

---

## Author Response (AR2)

*Response to the Reviewer 1*

R:The manuscript is ready for publication after correcting following points.
A: Thank you.

I found many careless mistakes, which must be fixed. Read and follow "Manuscript preparation guidances for authors" very carefully.
Done.

- Unit:
1. Spaces must be included between number and unit (e.g. 1 %, 1 m)
2. Units must be written exponentially (e.g. W m–2).
3. The metric system is mandatory and, wherever possible, SI units should be used. Also units in the denominator should be formatted with negative exponents (e.g. km h-1 instead of km/h)
Done.

- Unit should not be italicized.
Accounted for.

- Table: the word "Table" is never abbreviated and should be capitalized when followed by a number (e.g. Table 4).
Done.

- Cross referencing figure should be stated as "Fig. 1" not "Fig.1".
Check all cross referenced figures and tables in text.
Done.

- line 70: "~100 m" in stead of "sim100 m"?
- line 73: "it seems that..." instead of "in seems that..."?
Done.

- line 70: "TISL+CTMSL, ... in Fig. 3" but TISL+CTMSL is missed in Fig. 3.
TISL+CTMSL removed in accordance to the Fig.3 and former revision.

- There should be no indent for the paragraphs after equation (2) and (4)
- line 10: "Turbulence" in stead of "Turbulent" in the subsection header.
- line 30: Add "structure function" after "2nd-order"
- line 43: Italicize "u" and "v"
- line 96: Change "*" to "/times"
- line 76: Change "*" to "/times". Add one space before [m2/s] and fix the unit to m^-2 s.
line 3: Change "*" to "/times"
Everything corrected.

Figure 2
- Add more space between left and right panels.

- Italicize "u", "v", and "w" in caption.
Figure 3
- Use longer tick marks.
Figure 4
- Italicize "u", "v", and "w" in caption.
Figure 5
- Use longer tick marks.
- Italicize "u", "v", and "w" in caption.
Figure 6
- Use longer tick marks.
- Italicize "u", "v", and "w" in caption.
Figure 7
- Italicize "u", "v", and "w" in caption.
Figure 5
- Use longer tick marks.
All figures and captions corrected accordingly.

Table 1
- Italicize variables in caption.
Table 2
- Italicize variables in caption.
Table 3
- Fix unit to m^2 s^-3.
Table 4
- Add a space before unit in the header row.
All tables and captions corrected accordingly.

***Response to the Reviewer 2***

P3, L70 - the "sim" should be "~"
Corrected.

P6, L47 - should say "...can be somewhat…"
Corrected.

All changes in text are marked in the latexdiff file below.

[revised manuscript text omitted]

---

## Author Response (AR3)

Dear Editor,
we carefully came through the manuscript and corrected all mistakes you found and several other typos and imperfections. All changes introduced in the manuscript can be seen in the latexdiff file below.
Regards
Szymon Malinowski

[revised manuscript text omitted]

---

## Author Response (AR4)

Dear Editor,
several typos we found in the reference list are corrected. We checked guidelines to the authors
not finding a requirement to print titles in the references using italic font.
In a number of recent articles in ACP we could not find a single one using italics.
We can do this, but this is not required and seems to be not practised.
As it comes to doi numbers, older papers published by the American Meteorological Society
have such a "weird" ones, we verified them once more.
Regards
Szymon Malinowski

[revised manuscript text omitted]